**The glacial isostatic adjustment signal at present-day in northern Europe and the British Isles**
**estimated from geodetic observations and geophysical models**
Karen M. Simon[1*], Riccardo E.M. Riva[1], Marcel Kleinherenbrink[1], Thomas Frederikse[1,2]
[1]Delft University of Technology, Department of Geoscience and Remote Sensing, Stevinweg 1, 2628
CN Delft, the Netherlands
[2]Utrecht University, Institute for Marine and Atmospheric Research, Princetonplein 5, 3584 CC
Utrecht, the Netherlands
*Corresponding author: +31 15 2788147, k.m.simon@tudelft.nl
**Abstract**
The glacial isostatic adjustment (GIA) signal at present-day is constrained via joint inversion of
geodetic observations and GIA models for a region encompassing northern Europe, the British Isles,
and the Barents Sea. The constraining data are Global Positioning System (GPS) vertical crustal
velocities and GRACE (Gravity Recovery and Climate Experiment) gravity data. When the data are
inverted with a set of GIA models, the best-fit model for the vertical motion signal has a $\chi^2$ value of
approximately 1 and a maximum *a posteriori* uncertainty of 0.3-0.4 mm/yr. An elastic correction is
applied to the vertical land motion rates that accounts for present-day changes to terrestrial hydrology
as well as recent mass changes of ice sheets and glaciered regions. Throughout the study area, mass
losses from Greenland dominate the elastic vertical signal and combine to give an elastic correction of
up to +0.5 mm/yr in central Scandinavia. Neglecting to use an elastic correction may thus introduce a
small but persistent bias in model predictions of GIA vertical motion even in central Scandinavia where
vertical motion is dominated by GIA due to past glaciations. The predicted gravity signal is generally
less well-constrained than the vertical signal, in part due to uncertainties associated with the correction
for contemporary ice mass loss in Svalbard and the Russian Arctic. The GRACE-derived gravity trend
is corrected for present-day ice mass loss using estimates derived from the ICESat and CryoSat
missions, although a difference in magnitude between GRACE-inferred and altimetry-inferred regional
mass loss rates suggests the possibility of a non-negligible GIA response here either from millennial-
scale or Little Ice Age GIA.

## 1. Introduction

Glacial isostatic adjustment (GIA) is the process by which the Earth's crust and underlying mantle deform in response to surface loading and unloading by large ice sheets and glaciers (e.g., Peltier and Andrews 1976, Wu and Peltier 1982). Glacial isostatic deformation at present-day can include contributions from both recent (annual, decadal) variations to ice cover as well as contributions from millennial-scale variations in ice cover during Pleistocene and Holocene glaciation cycles, although in this study GIA refers to the latter paleo signal, specifically from the last glaciation. Ongoing GIA is usually the dominant present-day deformation signal in formerly glaciated areas (for example, up to approximately 1 cm/yr land uplift around the northwestern Gulf of Bothnia, Lidberg et al. 2010, Kierulf et al. 2014). Outside formerly glaciated regions, the GIA signal from past glaciations often remains large enough to form a significant component of observed present-day deformation and sea-level change rates. Constraint of the GIA signal at present-day is therefore required for accurate separation of the longer time scale and the more recent contributions to present-day land deformation and gravity change (Peltier 1998, Tamisiea 2011). This problem is complicated further by the fact that the GIA signal itself is temporally and spatially complex, therefore making it challenging for models to constrain some of the fundamental parameters relating to both ice cover during past glaciations and the structure of the Earth.

In Scandinavia, the GIA process has been studied extensively and constrained with data including relative sea level indicators, Global Positioning System (GPS) measurements and satellite gravity data (e.g., Lambeck et al. 1998, Milne et al. 2001, Steffen et al. 2010, see also Steffen and Wu (2011) for a review). While the GIA process in the region of the former Fennoscandian Ice Sheet is probably more extensively studied than anywhere else in the world, GIA in the Barents Sea is by comparison less well understood due in part to the lack of observational evidence left behind by a marine-based ice sheet. Auriac et al. (2016) provide a recent summary of GIA models in the Barents Sea region. Studies have also focussed on the smaller British Isles region, which experiences GIA deformation in response to deglaciation of both the local British Isles Ice Sheet and the larger adjacent Fennoscandian Ice Sheet (Bradley et al. 2011, Kuchar et al. 2012). The ice sheet evolution of the region as a whole was recently summarized by Patton et al. (2017). These studies and many others

have provided valuable insight into regional GIA processes. The majority of GIA models are however
forward models which can be limited by uncertainties in both the ice sheet model and Earth model.
Furthermore, because a best-fit forward GIA model is generally a single Earth-ice model combination,
their predictions of GIA deformations are typically provided without uncertainties.

This paper constrains the GIA signal in northern Europe through the simultaneous inversion of vertical
land motion rates from GPS and gravity change rates from GRACE (Gravity Recovery and Climate
Experiment). The semi-empirical method also estimates corresponding uncertainties for the preferred
model(s) which relative to forward model studies is a notable advantage of semi-empirical or data-
driven methodologies. Similar empirical and semi-empirical approaches have been implemented to
estimate regional long-term GIA signals in Antarctica (Riva et al. 2009, Gunter et al. 2014), North
America (Sasgen et al. 2012, Simon et al. 2017), Alaska (Jin et al. 2016) and Fennoscandia (Hill et al.
2010, Müller et al. 2012, Zhao et al. 2012). Here, our methodology is based on that of Hill et al. (2010);
relative to their previous work, we update both the GPS and GRACE datasets, incorporate a second
model ice sheet history into the *a priori* input, and expand the study area to include regions south and
west of Scandinavia, including the British Isles, as well as the Barents Sea to the north. Rather than
focus on model parameter estimation, we focus on constraint of the GIA signal at present-day. There
are three main goals: i) to model the paleo GIA signal at present-day in a continuous region between
Scandinavia and the British Isles, ii) to estimate empirically the uncertainty of the modelled signal, and
iii) to assess the importance of applying an elastic correction to the vertical land motion data.

**2. Model Inputs and Method**
2.1 GPS Data
Rates of vertical land motion measured by GPS are taken from both Kierulf et al. (2014) and the
Nevada Geodetic Laboratory (Blewitt et al. 2016) (**Figure 1**). The Kierulf et al. (2014) dataset has
relatively dense coverage within the region of the former load centre of the Fennoscandian Ice Sheet
(FIS), particularly in Norway, but sparse coverage elsewhere. The data from Blewitt et al. (2016) are
thus used for the region outside the former ice sheet margin. The Kierulf et al. (2014) dataset has 150
stations with time series lengths of at least 3 years. The data from Blewitt et al. (2016) span 1996-
2016 and have been limited to sites which have at least 10 years of data. To avoid spatial overlap of
sites, the data from Blewitt et al. (2016) have been additionally filtered to include only one site within a
30 km radius (where the site selected within the radius is the one with the largest number of usable
data epochs). The subset of data from Blewitt et al. (2016) has 309 stations. Combined with the Kierulf
et al. (2014) data, there are 459 measurements in total.

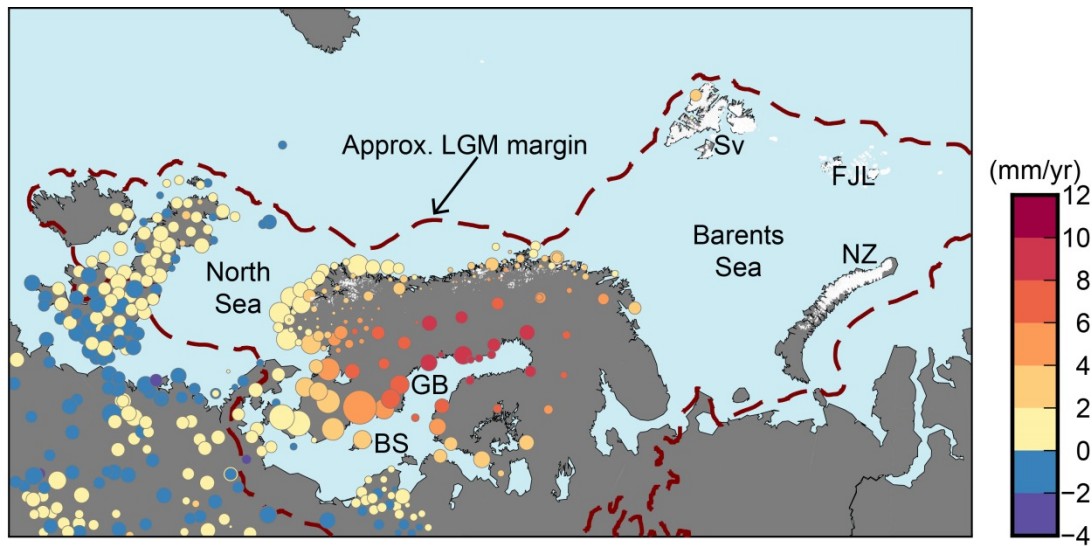


**Figure 1.** Rates of vertical land motion (mm/yr) for the GPS data used in the inversion, after correction
for elastic effects (Section 2.3). BS – Baltic Sea, FJL – Franz Josef Land, GB – Gulf of Bothnia, NZ –
Novaya Zemlya, Sv – Svalbard, FJL and NZ = Russian Arctic. Dark red dashed line (Hughes et al.
2016) shows the approximate boundary of ice cover at the Last Glacial Maximum (LGM) (ice cover on
Iceland not shown). White shading indicates present-day glaciers. The size of the circles is inversely
proportional to the measurement uncertainty.

As further described in Kierulf et al. (2014), their rates were derived using the GAMIT/GLOBK GPS
analysis software (Herring et al. 2011) and have uncertainties that assume a combination of white
noise and flicker noise, while the data from the Nevada Geodetic Laboratory were calculated using the
MIDAS trend estimator, an algorithm that is less sensitive to discontinuities in GPS time series (Blewitt
et al. 2016). Although the processing technique differs for each dataset, the two datasets are
combined in order to achieve the best possible spatial coverage in the study area. Common sites in
the two datasets compare within the observational uncertainties at all but two of thirty-one sites, and
no apparent bias is observed between the differences at the shared sites (**Figure A1**). Because the
uncertainties are consistently larger for the data from the Nevada Geodetic Laboratory than for the
data from Kierulf et al. (2014), we use the common sites to determine an average uncertainty scaling
factor (~2.25) to apply to the uncertainties in the latter dataset. The scaling avoids significantly biasing
the inversion result towards fitting either dataset. Both datasets are aligned in the International
Terrestrial Reference Frame 2008 (Altamimi et al. 2011), which is consistent with the CM frame to
within ~0.2 mm/yr. As described in Section 2.3, an elastic correction is applied that accounts for recent
changes in ice sheet and glacier volumes and terrestrial hydrology.

2.2 GRACE
The GRACE data are processed as in Simon et al. (2017). Rates of gravity change for a 10.5 year
period from 2004.02-2014.06 are estimated using 113 GRACE Release-05 (RL05) monthly solutions
from the University of Texas at Austin Center for Space Research (CSR). The coefficients are
truncated at degree and order 96. Part of the GIA signal may also be lost during the filtering,
particularly at higher orders; the typical spatial resolution of the signal is ~300 km (Siemes et al. 2013).
Values estimated from Satellite Laser Ranging (Cheng et al. 2013) replace the $C_{20}$ coefficients.
Following Klees et al. (2008), the monthly fields are filtered with a statistically optimal Wiener filter.
The optimal filter incorporates the full variance-covariance information of the monthly solutions, and
less aggressively filters in regions where signal is stronger. A mass trend is estimated that accounts
for bias, annual, and semi-annual variations (**Figure 2**). The signal uncertainty is represented by the
full variance-covariance matrix of the trend. Corrections for changes in the terrestrial hydrology cycle
and ice mass loss from Svalbard and the Russian Arctic are applied as described in Section 2.3.

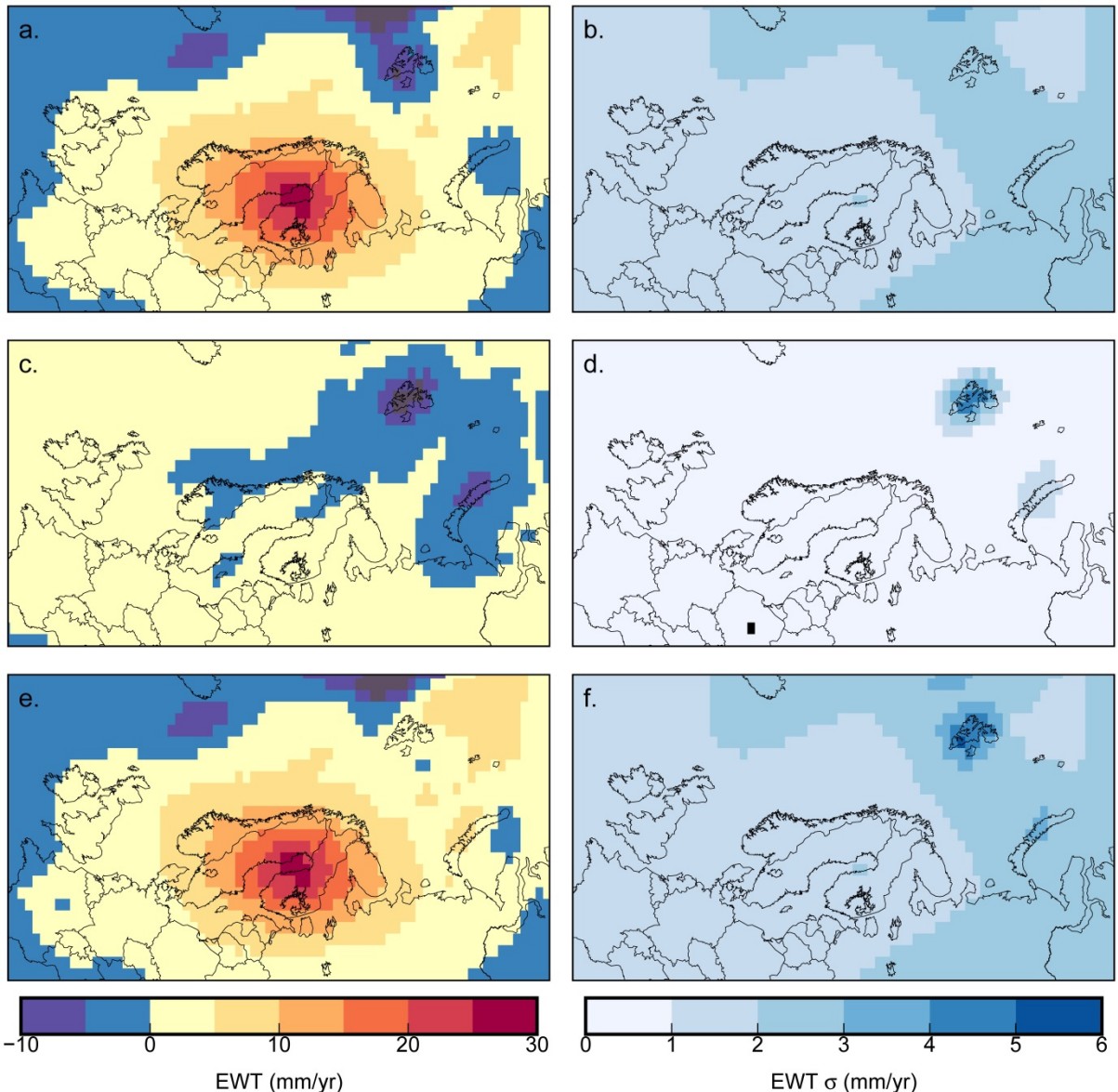

**Figure 2.** (a) Total gravity change rates measured from GRACE, (c) correction for terrestrial hydrology changes and present-day ice mass loss (Section 2.3), and (e) final corrected rates. (b,d,f) Same as (a,c,e) but rates are the $2\sigma$ uncertainties associated with the signal. Units are mm/yr change in equivalent water thickness (EWT).

2.3 Corrections for Terrestrial Hydrology and Present-day Ice Melt

Changes in terrestrial hydrology as well as present-day ice mass loss from Greenland, and glaciers

and ice caps in Svalbard, the Russian Arctic, and Scandinavia may form a significant contribution to

the total measured gravity change and vertical motion rates within the study area.

*GRACE*
In the continental region and south of approximately 71.5° N latitude, hydrological changes are the
sum of dam retention values (Chao et al. 2008) and anthropogenic groundwater depletion estimated
with the model PCR-GLOBWB (Wada et al. 2014). The trend is computed for 2004-2014 from 11
annual means on a 2° × 2° grid, consistent with the resolution of the GRACE data. In glaciered regions
(Scandinavia, Svalbard and the Russian Arctic), the hydrology model is not used to correct the input
rates. Rather, it is assumed that present-day estimates of regional ice melt derived from altimetry
observations should more accurately capture the dominant hydrological signals that would be
modelled by PCR-GLOBWB. The corrections for mass loss from the glaciers are also filtered to be
consistent with the spatial resolution of the GRACE data. The total correction for hydrology and glacial
mass loss is shown in **Figure 2c**, the individual contributions are shown in **Figure A2**.

Estimates of present-day mass changes in Scandinavia, the Russian Arctic, and Svalbard  are
summarized in **Table 1** for various studies, and vary considerably depending on estimation method
and time period. Ice mass loss in Scandinavia originates from glaciers in western Norway and is
consistently small with estimated rates between -1.2 to -2 Gt/yr. Here, we apply a mass loss rate of -
1.3 Gt/yr, determined by glaciological modelling (Marzeion et al. 2012, 2015).

In the Russian Arctic, glaciological estimates of mass change are consistent within uncertainties for
the different time periods and suggest mass change between -21.0 to -24.7 Gt/yr. These rates are
approximately twice those estimated by the ICESat and CryoSat missions, which estimate mass
changes in this region of between -10.5 to -14.9 Gt/yr, with a small acceleration observed after 2010
(Wouters, *pers. comm.*, 2016). The smallest net mass change estimate for the Russian Arctic comes
from GRACE, with -5.7 Gt/yr mass change observed between 2003-2013 (Schrama et al. 2014).

In Svalbard, estimated mass change rates are more discrepant. Again, glaciological estimates are the
largest, but two estimates of -42.0 Gt/yr and -17.0 Gt/yr between 2003-2009 are not consistent within
uncertainties and differ in magnitude by more than a factor of 2. Laser and radar altimetry estimates
are smaller, and suggest a clear acceleration in mass loss since 2010 (-4.6 Gt/yr between 2003-2009
and -16.5 Gt/yr between 2010-2014, Wouters, *pers. comm.*, 2016). As with the Russian Arctic,
GRACE is the estimation technique that records the smallest net mass change, with -4.0 Gt/yr
estimated in Svalbard between 2003-2013 (Schrama et al. 2014).

| Study/Source | Svalbard (Gt/yr) | Russian Arctic (Gt/yr) | Scandinavia (Gt/yr) |
|---|---|---|---|
| **2003-2009** | | | |
| Marzeion et al. (2015) *(2003-2009)* | -42.0 ± 3.2 (gl) | -22.9 ± 4.7 (gl) | -1.2 ± 0.2 (gl) |
| Gardner et al. (2013) *(2003-2009)* | -17.0 ± 6.0 (gl) -5.0 ± 2.0 (I, G) | -21.0 ± 13.0 (gl) -11.0 ± 4.0 (I, G) | -2.0 ± 0.0 (gl) |
| Wouters (2016) *(2003-2009)* | -4.6 ± 1.2 (I) | -10.5 ± 1.3 (I) | - |
| **2010-2014** | | | |
| Wouters (2016) *(2010-2014)* | -16.5 ± 1.6 (C) | -14.9 ± 1.2 (C) | - |
| **≥10 years time period** | | | |
| Marzeion et al. (2015) *(2004-2013)* | -39.8 ± 2.2 (gl) | -24.7 ± 3.0 (gl) | -1.3 ± 0.1 (gl) |
| Average Wouters (2016) *(2003-2014)* | -10.6 ± 2.0 (I, C) | -12.7 ± 1.8 (I, C) | - |
| Schrama et al. (2014) *(2003-2013)* | -4.0 ± 0.7 (G) | -5.7 ± 0.9 (G) | +1.3 ± 0.9 (G) |
| This study | -10.6 ± 2.0 (I, C) | -12.7 ± 1.8 (I, C) | -1.3 ± 0.1 (gl) |
| This study, with scaling | -2.7 ± 2.0 (I, C) | -2.5 ± 1.8 (I, C) | -1.3 ± 0.1 (gl)* |

**Table 1.** Estimates of present-day mass change for Svalbard, the Russian Arctic, and Scandinavia for
different time periods and from different sources. Letters in parentheses indicate estimation method; gl
- glaciological, I - IceSat, G - GRACE, C - CryoSat. All rates are in Gt/yr. *Not scaled.
GRACE measures total mass changes (solid Earth plus cryosphere), and thus a correction for one
needs to be applied in order to isolate the other. While the glaciological values and the altimetry
estimates (which are corrected for crustal uplift due to GIA) are both intended to represent changes to
the cryosphere, the differing mass change estimates among measurement techniques for the Russian
Arctic and Svalbard raise the question of which value to use when applying a correction to the total
GRACE trend shown in **Figure 2a**. Relative to GRACE, the glaciological and altimetry methods both
consistently infer larger mass losses, suggesting that GRACE contains a significant mass gain signal
from the solid Earth, either from glacial isostatic adjustment from the last glaciation, or from the Little
Ice Age (LIA). For both Svalbard and the Russian Arctic, we choose to apply an estimate that
averages the ICESat and CryoSat estimates over the years 2003-2014 (**Table 1**). Subtracting these
averaged rates from the total GRACE estimates for a similar time period (2003-2013, Schrama et al.
2014, **Table 1**), infers a reasonably consistent total solid Earth or GIA signal of +6.6-7 Gt/yr in the
region.

However, applying the averaged ice melt corrections to Svalbard and the Russian Arctic creates a
large mass gain signal over these two areas and a relatively smaller signal in the central Barents Sea;
this pattern is generally inconsistent with ice coverage in the Barents Sea region suggested by several
different Pleistocene ice sheet reconstructions (Auriac et al. 2016), and therefore inconsistent with the
paleo GIA signal that the input signal should represent. Possible explanations for this inconsistency
are: i) models of LGM ice cover in the region require thicker ice over Svalbard and the Russian Arctic
than in the Barents Sea, ii) there is a large Little Ice Age GIA signal over these two regions, and/or iii)
the Wiener filter applied to the GRACE data too aggressively filters signal in these small regions. The
first explanation is unlikely because glacial margin chronology suggests that Svalbard and the Russian
Arctic were located on or near the margin of the Barents Ice Sheet where ice cover would have been
thinnest. To counteract the effect of either of the latter two explanations (LIA rebound or signal loss in
GRACE), we apply ad-hoc scaling factors of 0.25 and 0.2 to the ice mass loss estimates in Svalbard
and the Russian Arctic (**Table 1**), so that their removal from the total GRACE signal results in a spatial
pattern in the residual (i.e., paleo GIA) signal that is approximately consistent with thicker LGM ice
cover over the Barents Sea than around its margins (**Figure 2e**). Such a scaling factor approach is
certainly not ideal, but serves to provide a GRACE input signal in the Barents Sea region that has a
spatial pattern broadly consistent with expectations of the paleo GIA response to loading and
unloading from the Barents Ice Sheet.
*GPS*
Vertical land motion rates may likewise be affected by present-day ice mass loss and the terrestrial
hydrology cycle. As with the GRACE data, the GPS data are corrected for changes to terrestrial
hydrology south of 71.5° N latitude using predictions from the PCR-GLOBWB model, although here,
the hydrology trend has been estimated from 1993-2014 to be more consistent with the length of the
GPS time series. North of 71.5° N latitude, the same scaled corrections derived from ICESat and
CryoSat are applied for present-day ice mass changes in Svalbard and the Russian Arctic.
Throughout the study area, the GPS measurements are also corrected for additional elastic vertical
motion from mass loss of the Greenland Ice Sheet, the Antarctic Ice Sheet and glaciers and ice caps
in northern Canada. Mass loss of the Greenland Ice Sheet is estimated from 1993-2014 using surface
mass balance estimates from RACMO2.3 (Noël et al. 2015) and ice discharge with a constant
acceleration of 6.6 Gt/yr$^2$ (van den Broeke et al. 2016). Mass loss of the Antarctic Ice Sheet is also
estimated from 1993-2014 using RACMO2.3p1 and assuming a constant acceleration in ice discharge
of 2 Gt/yr$^2$ (van Wessem et al. 2016). The scenarios for both Greenland and Antarctica are consistent
with the mass balance estimates from Shepherd et al. (2012). For the Canadian Arctic, a constant
mass loss rate of 60 Gt/yr is used (Gardner et al. 2013). All trends and accelerations are calculated
with annual time steps. The vertical elastic response is computed in the CM frame using a pseudo-
spectral approach up to degree and order 360 and includes the effect of rotational feedback. The
respective loads in each year are applied to a spherically symmetric Earth model (e.g., Farrell 1972)
using elastic Earth parameters from the Preliminary Reference Earth Model (Dziewonski and
Anderson 1981). Linear trends in the calculated vertical motion time series are then estimated by least
squares over the years 1993-2014 for each region, and finally summed to yield the total elastic
response. All signals combine to yield a total net uplift of approximately 0.2-0.5 mm/yr throughout most
of the study area, with Greenland mass loss providing the largest contribution (**Figure 3**). The
additional uncertainties are also computed and added in quadrature to the measurement uncertainties;
correction of the GPS data for non-GIA signals adds < ±0.05 mm/yr uncertainty in most of the study
area and ~±0.1 mm/yr in Svalbard (**Figure 3**).

Finally, in addition to present-day ice mass loss signals, a correction of 4.33 ± 0.40 mm/yr is removed
from the vertical motion rates for the two GPS sites on Svalbard (NYAL and LYRS). This value is an
average of 3 scenarios from Mémin et al. (2014) which estimate the vertical land motion at Ny-Ålesund
due to Pleistocene and Little Ice Age GIA signals; their estimates range from 3.31-4.95 mm/yr; thus
the averaged correction of 4.33 mm/yr that is applied assumes that the signal from Pleistocene GIA is
small and that most residual land motion here is from LIA rebound. After correction for present-day ice
mass changes and approximated LIA uplift, the residual (inferred paleo GIA) vertical uplift rates at
NYAL and LYRS are 2.64 ± 0.80 and 1.10 ± 2.64 mm/yr, respectively.

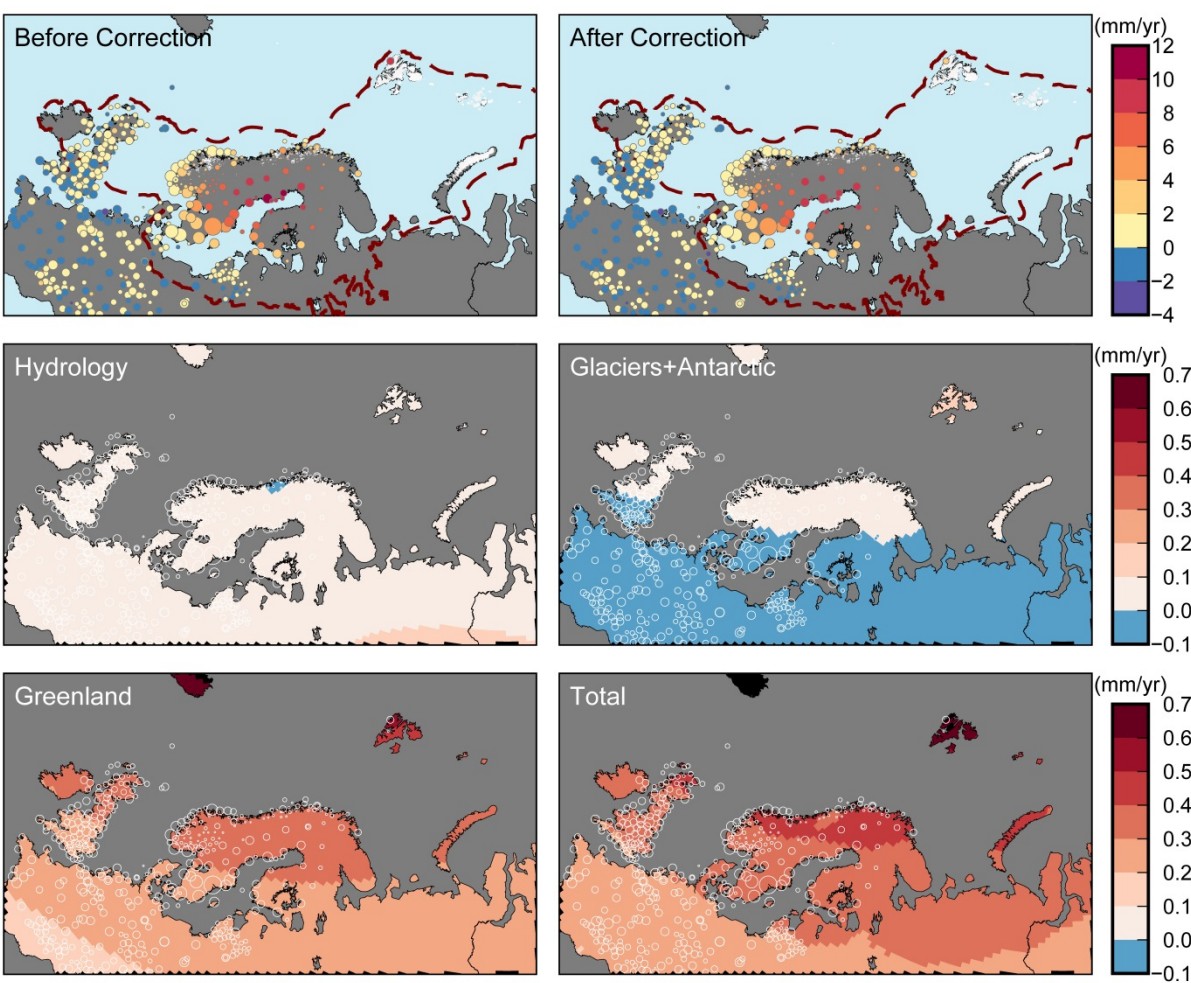

**Figure 3.** GPS-measured rates of vertical land motion before and after the applied elastic correction
(top left and right). An elastic correction is computed for mass loss from Greenland, the West Antarctic
Ice Sheet (WAIS), glaciers and ice caps in northern Canada, Svalbard and the Russian Arctic, and
loading from the terrestrial hydrology cycle. Sites on Svalbard are additionally corrected for LIA uplift
as discussed in the text.


2.4 *A Priori* Model Information
The prior model covariance matrix contains predictions from a set of forward GIA models that varies
ice sheet history and mantle viscosity and is constructed as described in Hill et al. (2010) and Simon
et al. (2017). Here, two different ice sheet histories are coupled to a suite of three-layer Earth models
with an elastic lithosphere and varying upper and lower mantle viscosities.

The first ice sheet model is the global ICE-5G model (Peltier 2004). We later compare the data-driven
predictions to the more recent ICE-6G forward model (Peltier et al. 2015) (Section 3.3); without ICE-
6G in the *a priori* information, the compared predictions are independent to the extent possible. In the
second ice sheet model, the glacial history over Fennoscandia and the British Isles is described by the
model(s) from the Australian National University (ANU, Lambeck et al. 2010). This second version of
the ice sheet model contains ICE-5G coverage over Greenland and Antarctica and the model of North
American coverage presented in Simon et al. (2015, 2016). Tests indicate that varying the ice sheet
history over North America has little impact on the predictions in Fennoscandia, although this variation
is useful for studies that wish to expand the study area outside of the current study area. Relative to
ICE-5G, LGM ice cover in the ANU model is thinner over the Barents Sea, thicker over Svalbard and
Scotland, and discontinuous between Scandinavia and the British Isles (**Figure 4**).

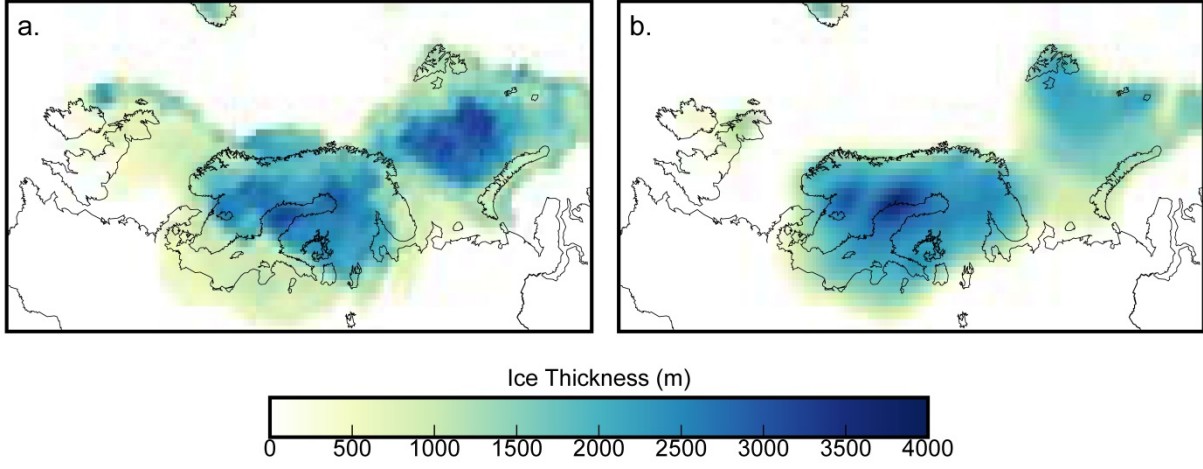

Ice Thickness (m)

0    500    1000    1500    2000    2500    3000    3500    4000


**Figure 4.** Last glacial maximum (LGM) ice cover in Scandinavia, the Barents Sea and the British Isles
from ICE-5G (a) and the ANU model (b).

Previous GIA modelling studies can be used to infer a range of reasonable Earth model parameters
for the *a priori* model set. Steffen and Wu (2011) reviewed the results of several GIA modelling studies
of the Fennoscandian region and indicated that these analyses suggest regional upper mantle
viscosities of between $0.1 - 1 \times 10^{21}$ Pa s and lower mantle viscosities approximately one to two
orders of magnitude larger (so $1 - 100 \times 10^{21}$ Pa s). They further indicated that lithospheric thickness
in Fennoscandia is likely variable with values ranging from 80 – 200 km (Steffen and Wu 2011).
Studies that have followed Steffen and Wu's (2011) review infer slightly narrower ranges for Earth
parameters in Fennoscandia. Depending on the ice sheet history and data constraints, the studies of
Zhao et al. (2012), Kierulf et al. (2014), Schmidt et al. (2014) and Patton et al. (2017) infer values of
upper mantle viscosity, lower mantle viscosity, and lithospheric thickness that may range from (or lie
within) $0.34 - 3 \times 10^{21}$ Pa s, $3 - 50 \times 10^{21}$ Pa s, and 93 – 160 km, respectively. In the British Isles,
Kuchar et al. (2012) infer upper and lower mantle viscosities of $3 \times 10^{21}$ Pa s and $2 \times 10^{22}$ Pa s
respectively, consistent with the values inferred by Bradley et al. (2011). Both studies find a best fit
lithospheric thickness of 71 km in this region. In the Barents Sea region, Auriac et al. (2016)
summarize the performance of six ice sheet models; the four best-fitting models infer respective upper
and lower mantle viscosities of $0.2 - 2 \times 10^{21}$ Pa s and $1 - 50 \times 10^{21}$ Pa s and lithospheric thicknesses
of 71 – 120 km. Both the studies of Root et al. (2015) and Patton et al. (2017) infer Earth parameters
for this region that are within the ranges given by Auriac et al. (2016).

Considering these three regions as a whole gives minimum to maximum ranges for upper and lower
mantle viscosity and lithospheric thickness of $0.2 - 3 \times 10^{21}$ Pa s, $3 - 50 \times 10^{21}$ Pa s and 71 – 160 km.
These mantle viscosity ranges are consistent with those used in our prior model set, which range from
$0.2 - 2 \times 10^{21}$ Pa s and $1 - 60 \times 10^{21}$ Pa s in the upper and lower mantle. The prior model set uses an
elastic lithospheric thickness of 90 km, although future analyses could benefit from use of a wider
range of thicknesses. With regard to the mantle viscosities, we note that both the ICE-5G and ANU ice
sheet models were not developed independently from a description of mantle viscosity. While the
coupling of a set of differing Earth models to a 'tuned' ice sheet history may introduce artificially high
variances, this concern may be countered by considering that the variances in such an *a priori* Earth-
ice model set could almost certainly be made larger if any combination of 3D Earth structure, non-
linear mantle rheology or glaciological and climatological constraints were additionally incorporated. A
full covariance matrix is generated that relates the variances of each model prediction relative to the
suite's average. All models are represented at spherical harmonic degree and order 256. The average
response and uncertainties of the *a priori* set is shown in **Figure 5**.

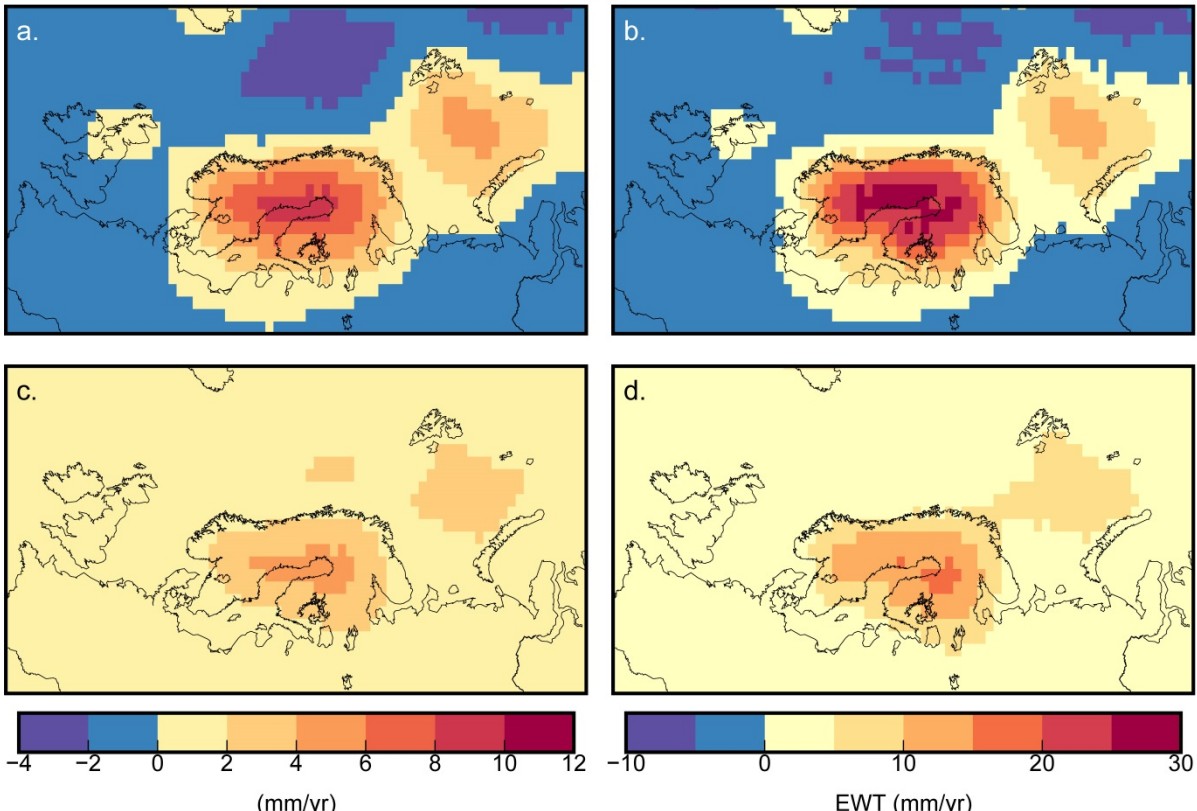

**Figure 5.** Averaged *a priori* rates of the Earth-ice model set. (a, c) Vertical rates and uncertainties. (b,
d) Gravity change rates and uncertainties in units of equivalent water thickness (EWT) change.

2.5 Method
The least-squares adjustment method is based on the methodology of Hill et al. (2010) and extended
by Simon et al. (2017). The method simultaneously inverts the data constraints (GPS, GRACE or
both) with the *a priori* GIA model information and minimizes the misfit to both input types. As in Simon
et al. (2017), variance component estimation (VCE) is also used to weight the input uncertainties. The
prior models are combined with the data in three scenarios: inversion with the GPS data alone (D1),
inversion with the GRACE data alone (D2), and inversion with both datasets (D3).

## 3. Results and Discussion

3.1 Prediction of Vertical Motion and Gravity Change

*Vertical Motion*

The predicted GIA response and uncertainties for the D1-D3 scenarios are shown for vertical land motion (**Figure 6**). The incorporation of the GPS data in scenarios D1 and D3 leads to a similar pattern of regional uplift although relative to D1, the D3 scenario predicts slightly lower rates of uplift over the northern British Isles and in the Barents Sea. D1 and D3 have respective peak uplift rates of 9.8 and 9.2 mm/yr. When only the gravity data are inverted in the D2 scenario, the region of uplift is broader and the peak uplift rate is smaller at 7.1 mm/yr. In all cases, the peak uplift is centred over the northwestern region of the Gulf of Bothnia. The peak ($1\sigma$) uncertainty rates are ±0.36, ±0.43 and ±0.28 mm/yr for the D1-D3 cases. Similar to the results of Simon et al. (2017), the predicted uncertainties are largest where the signal is largest (around the Gulf of Bothnia) and/or the data coverage is sparsest and most poorly constrained (around the Barents Sea). In Finland, for example, the relatively large signal and the relatively sparse data coverage combine to create a region of larger uncertainty than in surrounding areas. The inclusion of VCE does not significantly impact the signal prediction but in general somewhat increases the estimated *a posteriori* model uncertainty; the weighting factors determined by VCE are shown in **Table 2**. In model D1, both the uncertainties of the vertical velocities and the prior model set are slightly reduced. In model D3, the uncertainties of the vertical velocities are basically unscaled (increased by a factor of 1.02) whereas the covariances of the prior model set are reduced by a factor of 0.64 (note however that the original covariances of the prior model set are still generally larger than those of the vertical data, at least in the region of the former load centre).

*Gravity Change*

The predicted gravity change rates for D1-D3 are comparable to the predicted vertical motion rates in both the spatial pattern and relative magnitude (not shown). The peak mass change rates are again centred over the northern Gulf of Bothnia, and are 33.7, 24.3, and 32.3 mm/yr of equivalent water thickness change for the D1-D3 scenarios.  The peak associated $1\sigma$ uncertainties are ±1.59, ±1.59

and ±1.22 mm/yr EWT. In both the D2 and D3 models, the uncertainties of the GRACE data are
increased by the VCE analysis (**Table 2**).

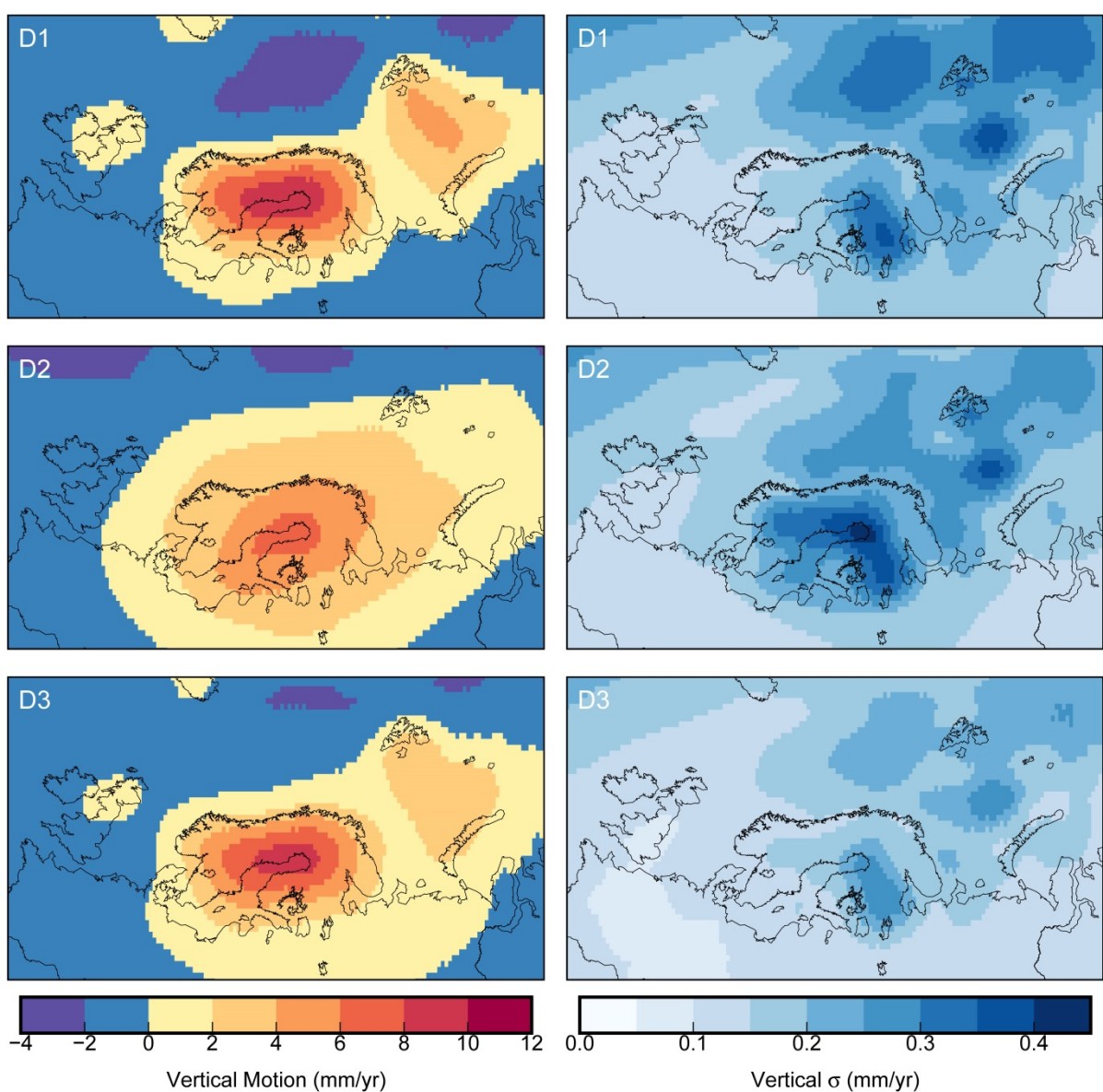


**Figure 6.** Prediction of present-day vertical land motion (left) and uncertainties (right) due to long-term
GIA for the D1-D3 scenarios.




| Data Incorporated | $\sigma^2$ Squared Value | | | Ratios | |
|---|---|---|---|---|---|
| | $\sigma_1^2$ (Vertical) | $\sigma_2^2$ (Gravity) | $\sigma_\mu^2$ (Prior) | $\sigma_1^2/\sigma_2^2$ | $\sigma_1^2/\sigma_\mu^2$, $\sigma_2^2/\sigma_\mu^2$ |
| D1: Vertical only | 0.85 | - | 0.94 | - | 0.90, - |
| D2: Gravity only | - | 13.51 | 0.61 | - | -, 22.15 |
| D3: Vertical+Gravity | 1.02 | 20.55 | 0.64 | 0.05 | 1.59, 32.11 |

**Table 2.** Results of the variance component analysis. $\sigma_1^2$ and $\sigma_2^2$ are the variance factors applied to the
vertical motion data (dataset 1) and gravity change data (dataset 2), respectively, and $\sigma_\mu^2$ is the
variance factor applied to the prior information. The ratios describe how each input covariance matrix
is weighted relative to the other(s).

3.2 Misfit Values and Residuals
For both $\chi^2$ and RMS values, the D1 model provides the best fit to the vertical data, the D2 model
provides the best fit to the gravity data, and the D3 model provides the best fit overall (**Figure 7**). The
$\chi^2$ values of the vertical prediction for both D1 and D3 are approximately equal to 1. The $\chi^2$ values for
the gravity data are relatively large with the smallest value of 15.9 obtained for the D2 model. Scaling
the gravity data uncertainties by the VCE-determined scaling factors in **Table 2** reduces the overall $\chi^2$
values for the gravity prediction to  approximately 1.2 for the D2 and D3 models. However, the
statistical fit of the models to the gravity data remains generally worse than the fit to the vertical motion
data.

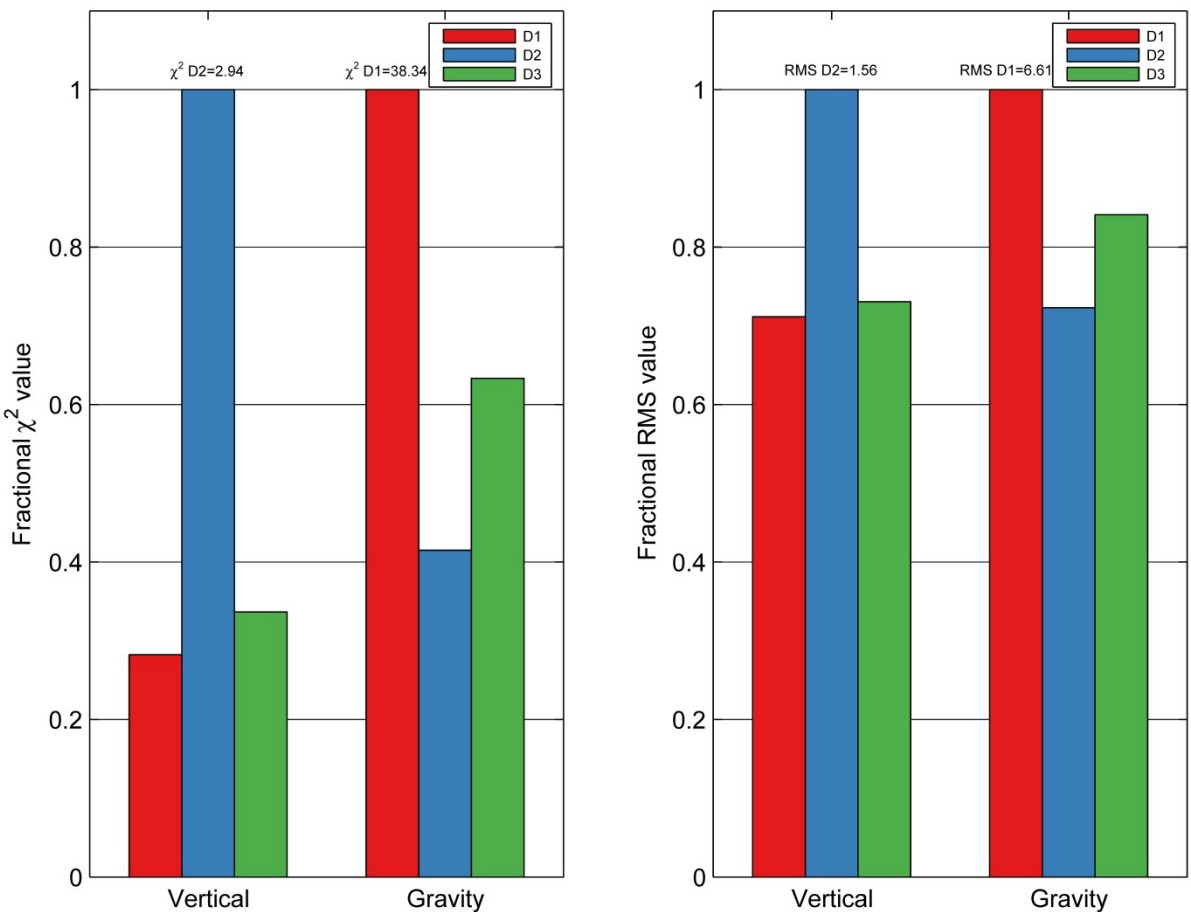


**Figure 7.** Fractional $\chi^2$ and RMS values for each of the D1-D3 models. Fractional values are determined relative to the value of the worst fitting model for both the vertical motion and gravity change predictions (i.e., fractional $\chi^2$ values of the vertical motion prediction are relative to D2 for which $\chi^2 = 2.94$). $\chi^2$ values are not VCE-scaled; see **Figure 8** for all $\chi^2$ values including with and without VCE scaling, where applicable.



**Figures 8-9** summarize the spatial residuals for the best-fit D3 model and the binned residuals for all
models. The vertical motion residuals are unbiased and generally small. Regionally, the D3 model
underpredicts vertical motion in Scotland and conversely overpredicts vertical motion along parts of
the southern Norwegian coast and the Netherlands. The gravity residuals for D3 are relatively low for
much of the study area, although there is noticeable overprediction in central Scandinavia and in the
Barents Sea.

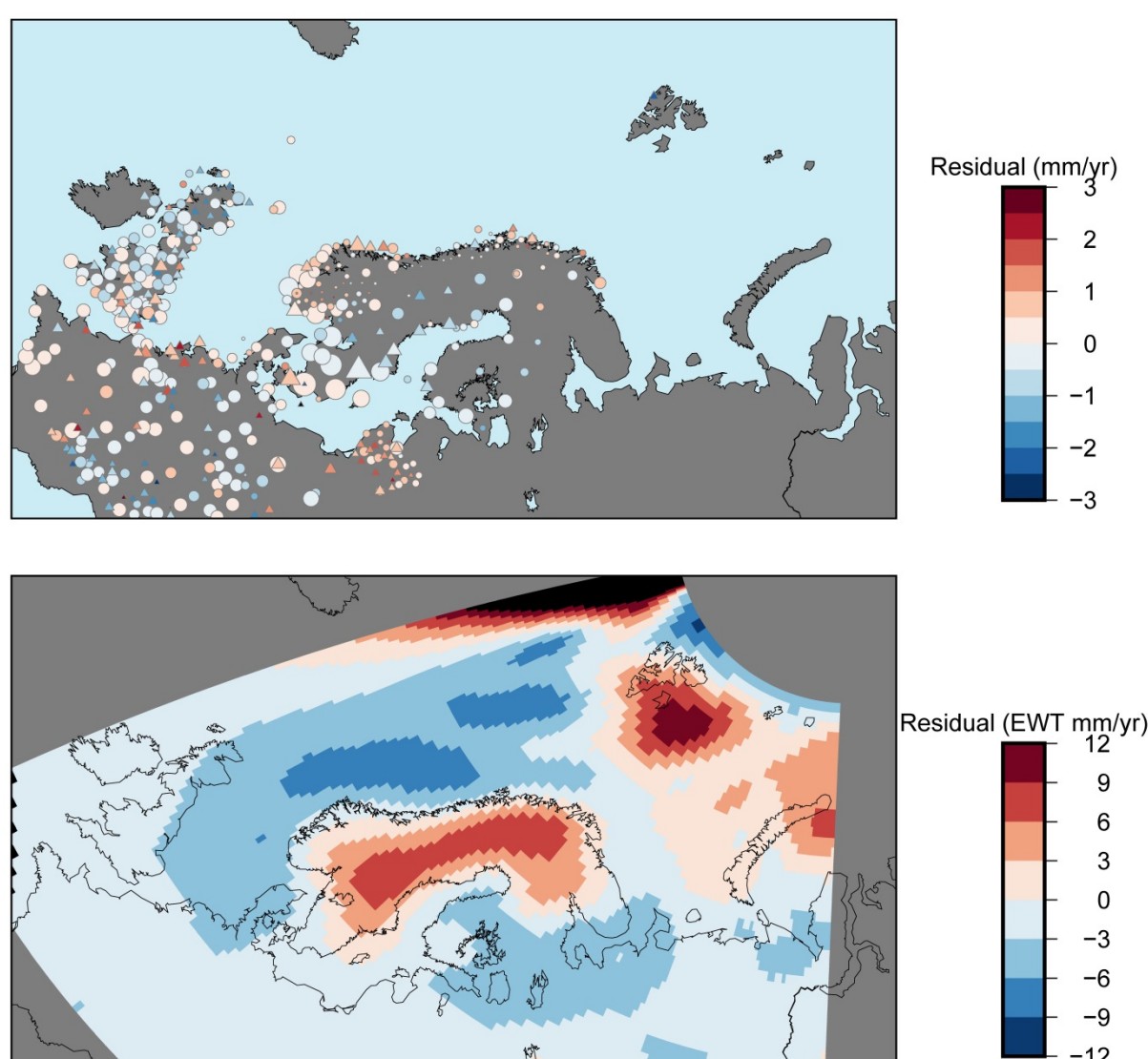

**Figure 8.** Spatial residuals for the D3 model for vertical motion (top) and gravity change (bottom). In top panel, triangles indicate model prediction is outside the 1σ uncertainty of the measurement, circles indicate model prediction is inside the 1σ uncertainty of the measurement.

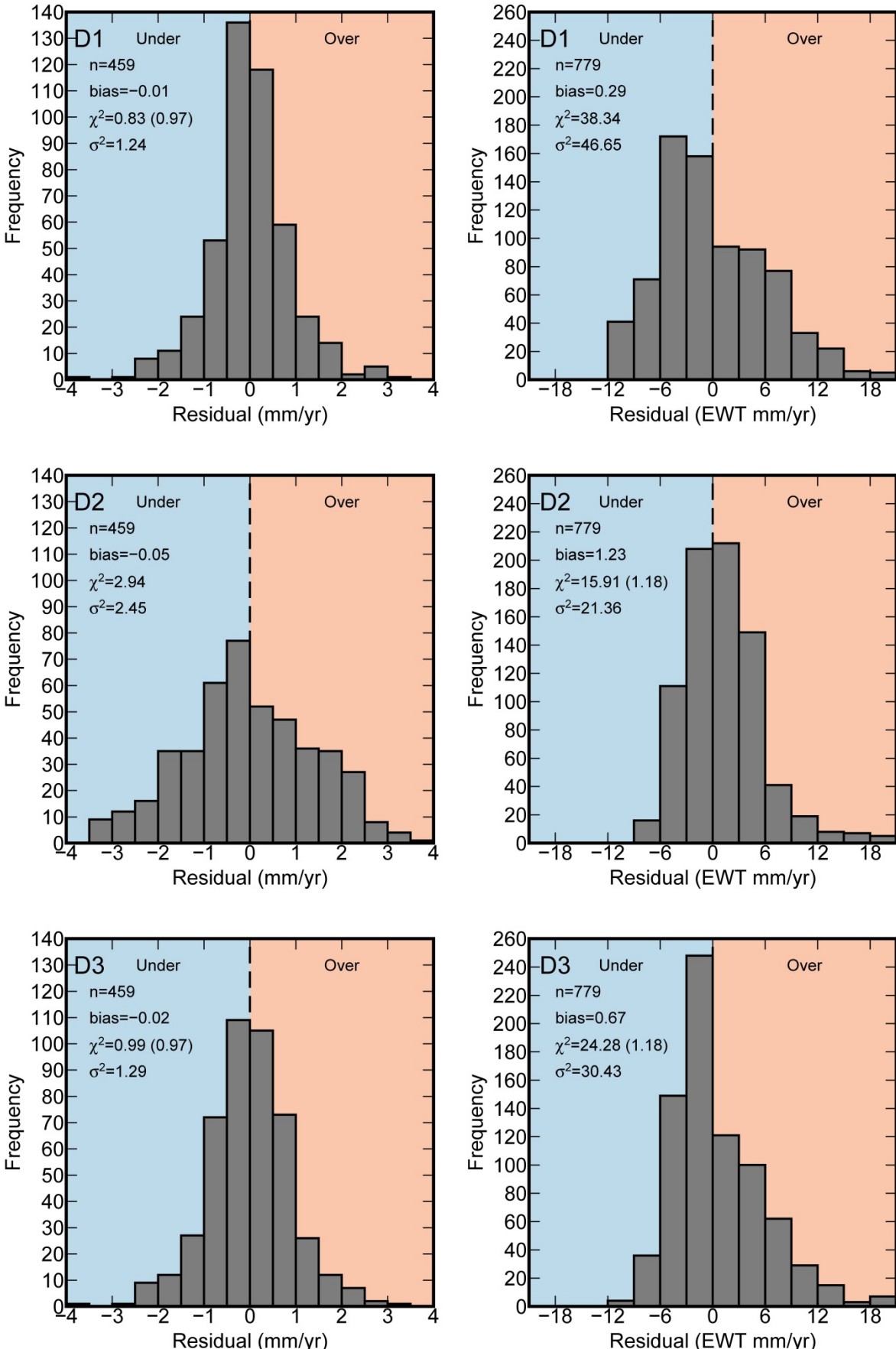

**Figure 9.** Histogram of residuals for models D1-D3, for prediction of vertical motion (left) and gravity
change (right). Pink and blue shading indicate model overprediction and underprediction, respectively.
Where given, $\chi^2$ values in brackets show the VCE-scaled $\chi^2$ value.


3.3 Comparison of Vertical Motion Prediction to Other Models
We compare the vertical motion prediction of D1 to two other models. The first model is the forward
GIA model ICE-6G (Peltier et al. 2015) which is constrained by a global dataset of vertical land motion
measurements. The majority of the these data are GPS measurements from the global solution of
JPL; within the study area of Scandinavia and northern Europe, additional measurements come from
the BIFROST GPS network as well as a small number of SLR, DORIS and VLBI measurements
(Argus et al. 2014, Peltier et al. 2015). The second model is the semi-empirical land uplift model
NKG2016LU (Vestøl et al. 2016) designed by several researchers in collaboration with the Nordic
Geodetic Commission (NKG). This model is constrained with GPS-measured vertical land motion
rates updated from the dataset of Kierulf et al. (2014), levelling measurements and GIA model
predictions and provides a semi-empirical estimate of total present-day vertical land motion.

**Figure 10** compares the vertical land motion predictions of D1, ICE-6G and NKG2016LU. The ICE-6G
comparison is made relative to the vertical motion dataset presented in this paper, although as stated
above, it was constrained with a different variant of regional vertical land motion data. As well,
NKG2016LU predictions are available on a smaller grid and best fits data from Scandinavia and the
Baltic countries, thus, we limit our comparison with this model to north of 55°N (reducing the
comparison dataset from 459 to 185 sites).

With no significant bias and a $\chi^2$ value of less than 1, the D1 model provides a good fit to the data. As
with the D3 model, the D1 model underpredicts vertical motion over the northern British Isles, and
appears also to overpredict vertical motion around the Netherlands. The ICE-6G model underpredicts
vertical motion at several sites in Scandinavia and has an overall $\chi^2$ value of 1.33, somewhat higher
than that of D1. At station NYAL on Svalbard, both the D1 and ICE-6G models underpredict vertical
motion by more than 2 mm/yr, even after the applied corrections for present-day mass loss and
possible LIA uplift. When the NKG2016LU model is evaluated relative to the GPS data without an
elastic correction applied, the $\chi^2$ value is less than 1, similar to D1. **Figure 10** shows the difference in
the prediction of vertical motion between NKG2016LU and D1. The former has consistently higher
predicted uplift rates over the study area, with an average difference of +0.3 mm/yr., which is primarily
the result of applying the elastic correction to the data used in the D1 model. D1 is therefore to the
extent that is possible, an estimate of the paleo GIA signal rather than the total uplift signal. That the
statistical fit to the data of both D1 and NKG2016LU is slightly better than the fit of the ICE-6G forward
model is expected due to the fundamental difference in model type: unlike ICE-6G, both of the semi-
empirical models explicitly incorporate the data into the prediction via formal inversion. Conversely, an
advantage of ICE-6G and other models of its type is the direct insight they offer into the space-time
evolution of the ice sheets, which cannot be inferred from a present-day empirical prediction alone.

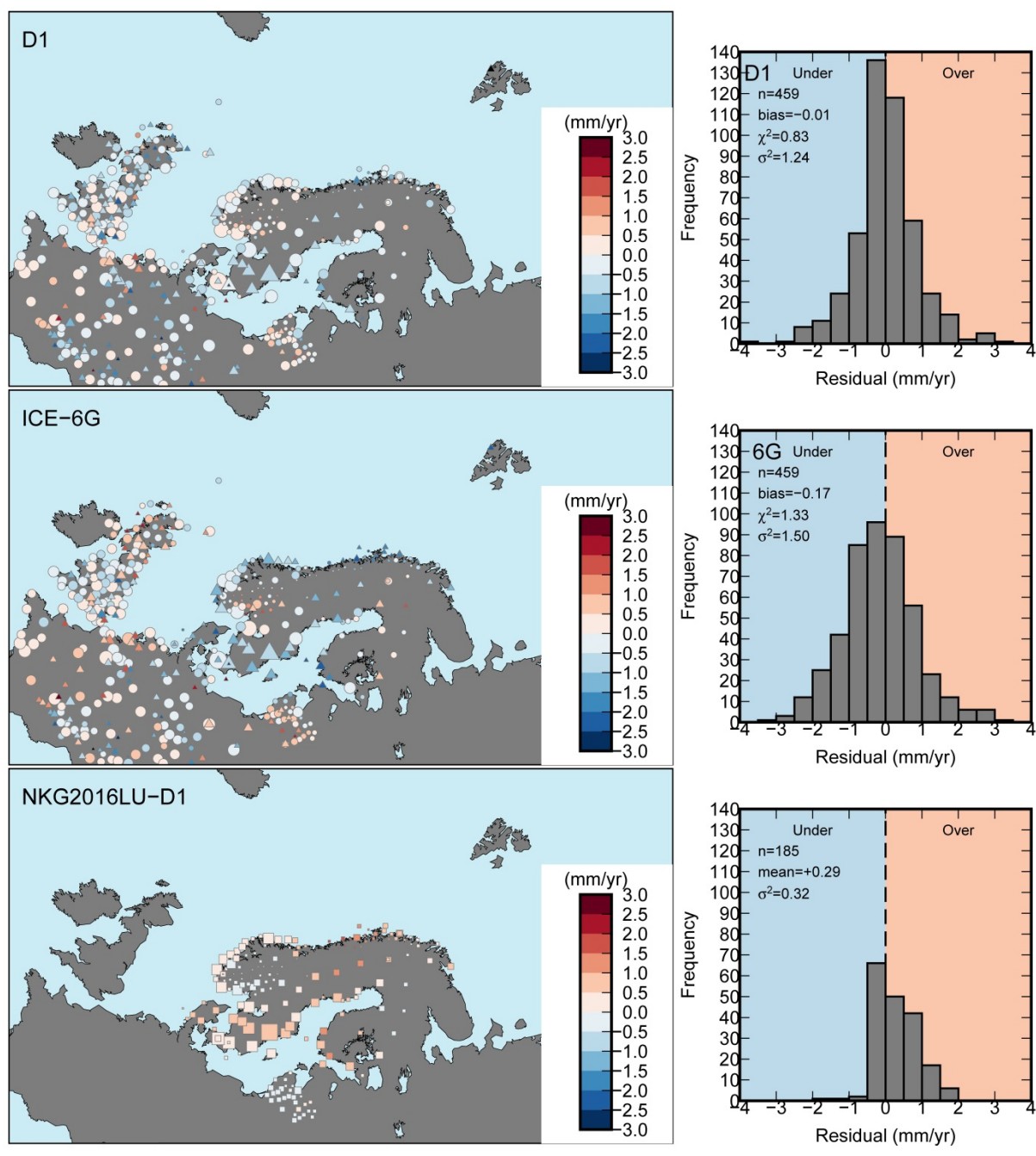


**Figure 10.** Spatial (left) and binned (right) vertical motion residuals for D1 and ICE-6G and the
difference between the NKG2016LU and D1 models. Triangles indicate model prediction is outside the
1σ uncertainty of the measurement, circles indicate model prediction is inside the 1σ uncertainty of the
measurement, squares show the difference between the two models (bottom left).


3.4 Tide Gauge Comparison
To assess the effect of GIA on regional sea-level change, we remove model D1's predictions of long-
term GIA from mean sea-level trends at 13 tide gauge sites along the coast of the North Sea and 7
tide gauge sites along the Norwegian coast (**Figures 11, 12**). The sea-level trends are taken from
Frederikse et al. (2016) who estimated the rates at Permanent Service for Mean Sea Level (PSMSL)
sites over the time interval 1958-2014. We also compare the effect of removing the modelled relative
sea-level rates of ICE-6G at the same PSMSL locations. For both the North Sea and the Norwegian
coastline, application of the D1 long-term sea-level trends to the total sea-level trends reduces the
interstation variability and infers a similar rate of non-GIA sea-level change (1.89 mm/yr and 1.84
mm/yr respectively).

*North Sea*
When corrected for the D1 long-term GIA trends, which are assumed to be linear over decadal time-
scales, the standard deviation of the trends decreases somewhat from 0.81 mm/yr to 0.71 mm/yr. The
D1 GIA correction is small at most sites, and at all sites except 7-9 (Hirtshals, Tregde and Stavanger),
the averaged sea-level trends appear dominated by processes other than long-term GIA (**Figure 11**).
At Hirtshals, Tregde and Stavanger, which are located nearest to the centre of the former FIS, the
predicted GIA-induced sea-level trend is more than twice the magnitude of the averaged sea-level
trend and removing the GIA signal shifts the original trend at these locations closer to the mean of the
13 locations. When the ICE-6G rates are removed from the sea-level trends, the interstation variability
and standard deviation (from 0.81 mm/yr to 0.83 mm/yr) are relatively unchanged. Regionally, the
average D1 GIA model trend is ~-0.45 mm/yr for the North Sea which is larger in magnitude than the
ICE-6G GIA trend of ~0.06 mm/yr in the North Sea. This difference may in part be due to the influence
of the ANU ice sheet model in the prior model, which predicts stronger subsidence over the North Sea
than either ICE-5G or ICE-6G. Accordingly, removal of the GIA signal from all 13 locations changes
the North Sea mean sea-level trend from 1.39 mm/yr to 1.84 mm/yr for D1 and to 1.33 mm/yr for ICE-
6G. Station Lerwick is particularly discrepant; removing it from the comparison decreases the standard
deviation of the non-GIA rates to 0.45 mm/yr for D1 and 0.75 mm/yr for ICE-6G. The variability at
Lerwick is insensitive to application of the relatively small and linear GIA correction for this region and
cannot be explained by GIA-induced sea-level change. Conversely, the variability in sea-level trends
in the northeast North Sea, near the former FIS, is easily attributed to GIA for model D1.

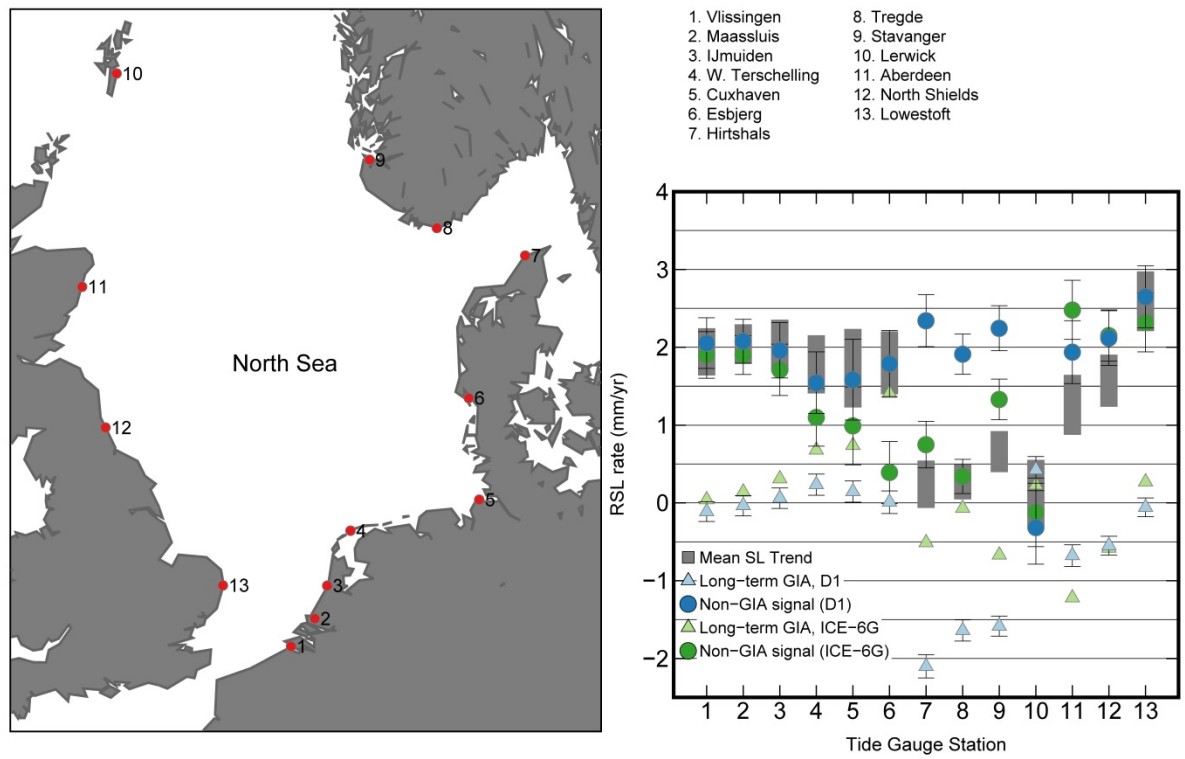

**Figure 11.** Comparison of mean total, long-term GIA and non-GIA sea-level trends (grey boxes,
triangles, circles) for 13 tide gauge stations in the North Sea. Long-term GIA trends are from model D1
and ICE-6G, mean sea-level trends are from Frederikse et al. (2016).

*Norwegian Coast*
The average sea-level trend for the 7 sites along the Norwegian coast is -0.22 mm/yr with a standard
deviation of 0.87 mm/yr. Removal of the D1 long-term GIA trends increases the average sea-level
trend to 1.89 mm/yr and reduces the interstation variability (0.44 mm/yr standard deviation) (**Figure**
**12**). The same is true for ICE-6G, although the magnitude of the changes are smaller (0.44 mm/yr
mean, 0.65 mm/yr standard deviation). This difference is owing to the relatively larger average GIA-
related relative sea-level change for D1 (-2.11 mm/yr) compared to ICE-6G (-0.66 mm/yr). The
gradient of predicted GIA changes across the Norwegian coastline is steep, so the results may also be
sensitive to the resolution of the GIA models.

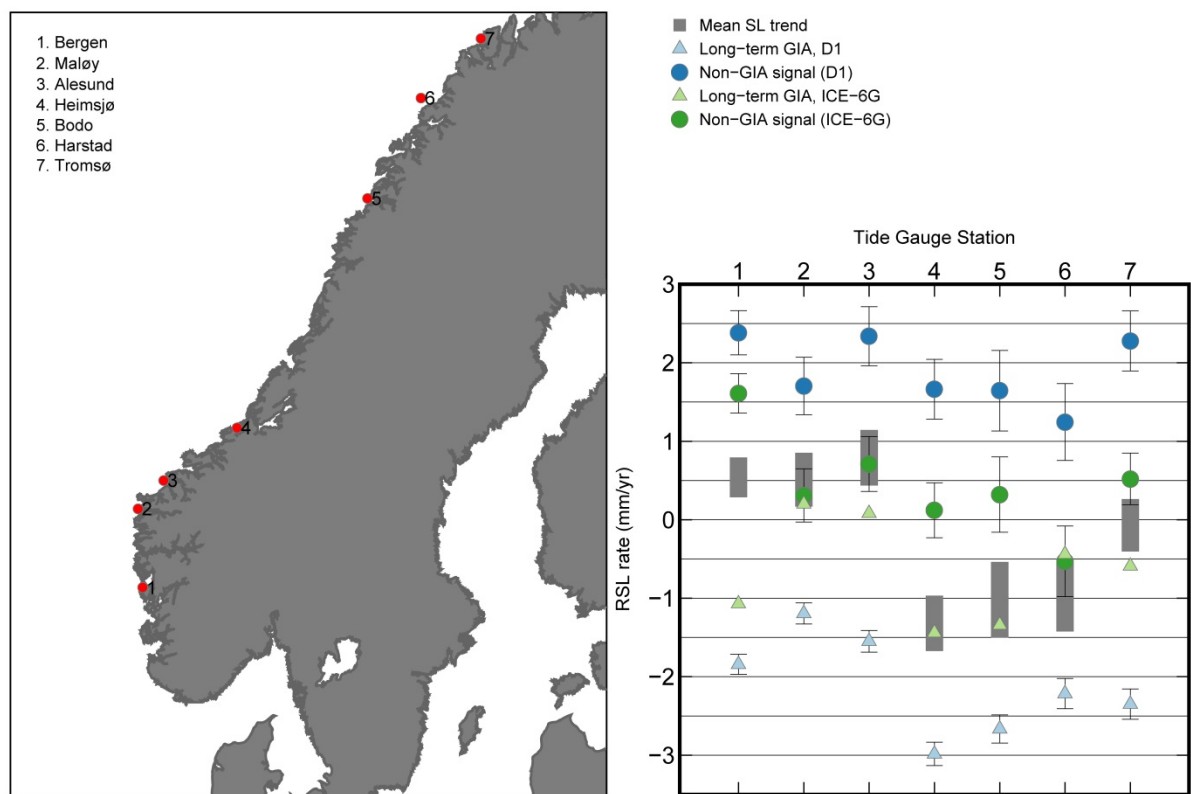

**Figure 12.** Same as caption for Figure 11, except for tide gauge locations along the Norwegian
coastline.

**4. Conclusion**
We generate a data-driven prediction of the long-term GIA response at present-day in Scandinavia,
northern Europe and the Barents Sea through the simultaneous inversion of GPS-measured vertical
motion rates, GRACE-measured gravity change rates, and *a priori* GIA model information. In models
D1-D3, we predict GIA motions for the inversion of the vertical motion data, the gravity data, and both
datasets. In both the $\chi^2$ and RMS sense, the vertical motion data alone have the poorest ability to
predict gravity change, and vice versa. Predictions of the D3 model provide the best overall fit to both
datasets.

In general, prediction of the gravity signal is problematic, with larger $\chi^2$ values than those obtained for
the vertical motion prediction. The poorer prediction of gravity change is in part due to the uncertainty
of the present-day mass loss effect in the Barents Sea region. The mass loss signal estimated by
GRACE over Svalbard and the Russian Arctic is significantly smaller than estimates obtained from
satellite altimetry. This difference may be the result of signal loss in the GRACE data from application
of the Wiener filter or may also indicate that there is a non-zero component of ongoing glacial isostatic
adjustment from the LIA.

The vertical motion signal is overall better predicted than the gravity signal. Both the D1 and D3
models have $\chi^2$ values of ≤ 1 and predict rates of vertical motion that are within the 1σ uncertainty of
the observations throughout most of the study area. Regions of misfit persist in Scotland and around
the Netherlands, where the model underpredicts and overpredicts rates of vertical motion,
respectively. The misfit in Scotland may be partly due to both positive and negative rates of vertical
motion that are present in the data over relatively short distances. Further analysis and filtering of the
GPS dataset may be useful in this region. In the Netherlands, Kooi et al. (1998) found that present-day
subsidence from sediment compaction as well as tectonic movements may contribute significantly to
vertical land motion; correction for these effects may serve to reduce some of the residuals in this
region. There may also be significant neotectonic movements in central Norway (Kierulf et al. 2014),
which may explain some of the misfits that remain mainly along the central Norwegian coastline
(**Figure 8**).

The prediction of vertical land motion has a small but non-negligible sensitivity to the application of an
elastic correction. The elastic correction applied in this study is between 0.2-0.5 mm/yr; the largest
contribution comes from mass loss of the Greenland Ice Sheet which yields regional uplift with a
southeastward decreasing gradient. When the model predictions from another semi-empirical model of
vertical motion, NKG2016LU, are compared to D1, a small but relatively uniform difference of +0.3
mm/yr is present in the model predictions over Scandinavia. Both NKG2016LU and D1 (and D3) have
vertical motion $\chi^2$ values ≤ 1 over their respective study areas. However, while the magnitude of the
difference is smaller than the observational uncertainty on many of the measurements, it is generally
larger than the estimated *a posteriori* model uncertainty. Also, because only anthropogenic
hydrological signals (and not natural hydrological signals) were included in the elastic correction, it is
possible that the applied elastic correction is conservative in this region.

Therefore, the presence of such a difference in the vertical motion prediction suggests that while long-
term GIA is the dominant contributor to vertical motion in central Scandinavia, that it is still worthwhile
to correct GPS land motion rates for present-day elastic signals, so long as these signals are
adequately approximated (e.g., Riva et al. 2017). This conclusion however highlights a fundamental
assumption that underpins the data-driven methodology: that the input data can be adequately
'cleaned' for processes not arising from long-term GIA. Even with applied corrections for hydrology
and contemporary ice mass loss, this assumption may not always be adequate, especially in regions
where model misfits relative to the data are spatially coherent. Thus, the success of data-driven GIA
predictions are evaluated by two criteria: i) the estimation of realistic *a posteriori* uncertainties that are
smaller than those associated with *a priori* knowledge and measurement uncertainty, and ii) the ability
of the final model to provide a good fit to the data. The vertical motion predictions of models D1 and
D3 satisfy both criteria for most of the study area and thus can provide a useful tool with which to
separate long-term GIA signals from shorter-term forcing.

**Data Availability**

Gridded vertical land motion predictions for the D1 model are available at the 4TU Centre for
Research Data repository, https://data.4tu.nl/, doi:10.4121/uuid:4a495bbc-0478-483a-baef-
19ff34103dd2.

 **Appendix**

The 31 GPS measurements that are common to the Kierulf et al. (2014) and Nevada Geodetic
Laboratory (Blewitt et al. 2016) datasets are shown in **Figure A1**. The individual anthropogenic
hydrology and glacial mass change contributions to the GRACE correction are shown in **Figure A2.**

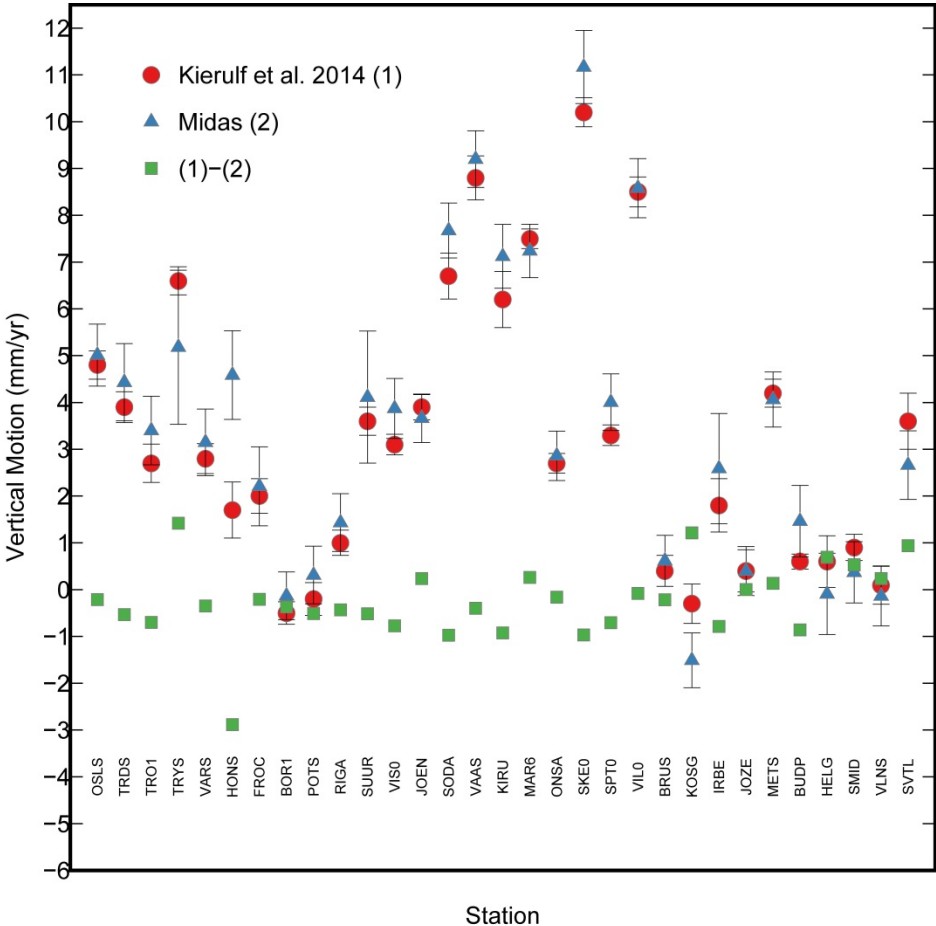


**Figure A1.** Vertical land motion measurements at 31 sites common to both datasets used in this
study.

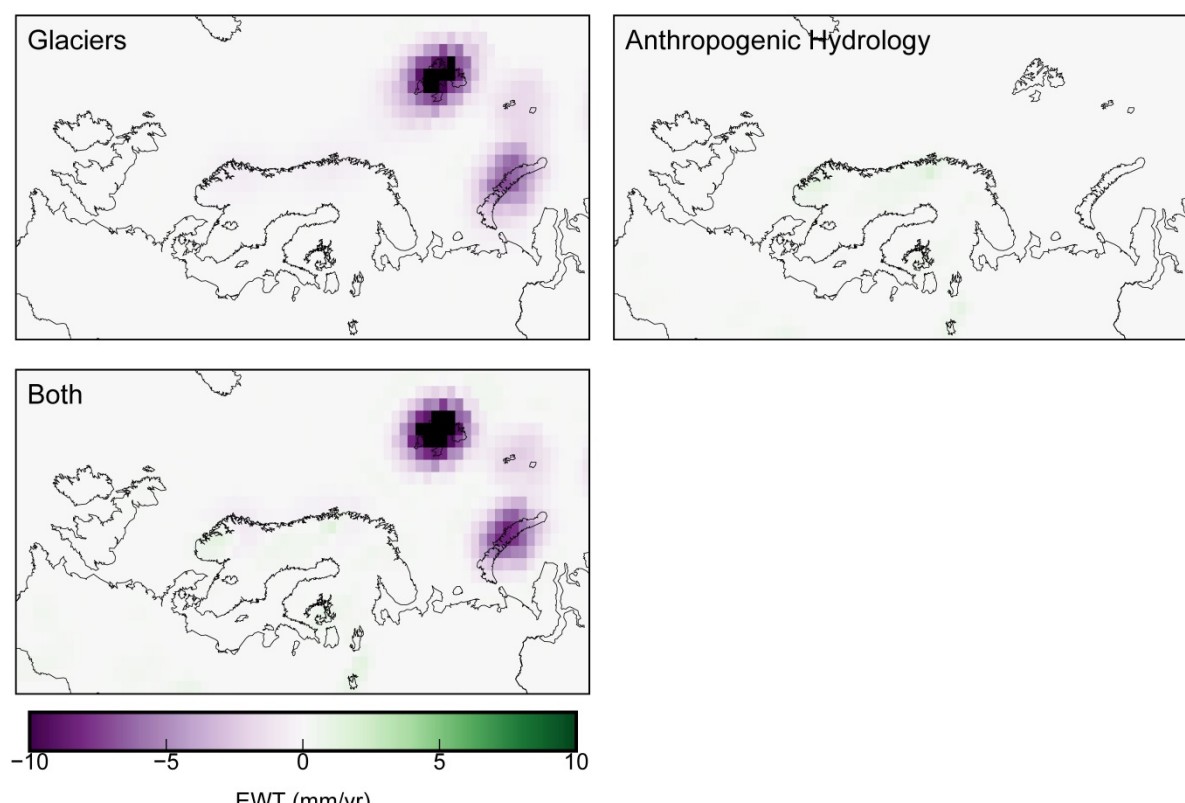


**Figure A2.** Individual and combined contributions to the correction applied to the GRACE data (combined is the same as **Figure 2c**).



**Acknowledgements**
We would like to thank Anthony Purcell for providing the ANU ice sheet model for Europe and the
British Isles, Yoshihide Wada for making the PCR-GLOBWB hydrology model available, and Bert
Wouters for providing altimetry estimates of recent mass loss for Svalbard and the Russian Arctic. We
also thank two anonymous reviewers for comments that improved the manuscript. This work is part of
the project for a Multi-Scale Sea-Level model (MuSSeL), funded by the Netherlands Organization for
Scientific Research, VIDI Grant No. 864.12.012.

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
