# Peer review of "The glacial isostatic adjustment signal at present-day in northern Europe and the British Isles"

_Solid Earth, 2018_

## Referee Comment (RC1) · Anonymous Referee #1 · 8 Mar 2018

**General comments**

Simon et al. present a semi-empirical model of the glacial isostatic adjustment (GIA) signal in northern Europe, the British Isles and the Barents Sea. It is generated in a least-square adjustment method with the help of data of the Global Navigation Satellite System (GNSS) and the Gravity Recovery and Climate Experiment (GRACE), and additional input from GIA models. It is the first such model for this large area and is assumed to represent the GIA signal mainly induced by the last glaciation. The method has been used before by the main author and Hill et al. (2010) but for other or smaller areas. Data sets with longer time spans and interesting corrections are used to provide

a model with realistic uncertainties, which are missing for most GIA models. Finally, the model is compared to other models and its performance for correction of tide-gauge observations in the North Sea is tested.

In principle, I would recommend publication of this study. However, there are many points and suggestions below the authors should work on before I would give my thumbs up. They are at a level where I suggest major revision as some reading and rephrasing is necessary. Despite the English is fine, the figures have high quality and the paper reads well at first glance, I identified some sloppiness and to be frank, the acknowledgement of related works on GIA in the investigated area is at a very low level. It appears to me that the authors consider many facts as known to everyone and thus refrain from explaining abbreviations and referencing whole paragraphs. The authors should also more discuss the reliability of GRACE in semi-empirical GIA models. The weakest part of the study is tuning GRACE results to fit expectations, and in the end the model is to less than 5% constrained by GRACE. It would be nice to read a discussion on how important these less than 5% are.

**Specific comments**

Title: I have four issues, (1) Scandinavia is part of northern Europe but you miss to mention the Barents Sea, which is largely discussed in your manuscript, (2) abbreviations should not be used in the title except they are well known, (3) although I understand that you want to distinguish the GIA signal you investigate from current climate change-induced GIA signals, "long-term", i.e. its definition, is not the best word to me (see below), and (4) you estimate your signal also with help of GIA models. My suggestion would be: The glacial isostatic adjustment signal in northern Europe, the British Isles and the Barents Sea estimated from satellite positioning and space gravimetry data, and geophysical modelling

L11/L29-31: it seems "long-term" refers to "ice sheets... during the last glaciation" in your introductory lines 29-31. However, parts of this signal can result from previous

glaciations, see e.g., Johnston & Lambeck (1999) and Root et al. (2015). This should be either specified or the word "long-term" be dropped.

L12: suggest change GPS to GNSS and introduce abbreviation; GPS is one of the GNSSs like Galileo, GLONASS or BeiDou. You can specify in the main text that both Kierulf et al. and Blewitt et al. use GPS only.

L12: explain GRACE abbreviation

L12: delete "Scandinavia," (or do you mean northern Central Europe? – but my suggestion would be simply "northern Europe" which includes Scandinavia)

Introduction, i.e. L29-40: this is a rather short introduction that combines a paragraph without any references to a paragraph with references but already specifically focussed on the paper's topic. I suggest mention a few "early" general studies on GIA in the first paragraph, e.g. Peltier & Andrews (1976) and Wu & Peltier (1982). Otherwise it sounds that GIA should be well known for the reader. A reference for the 1 cm/a and the location should be added. I am aware that GIA in Fennoscandia has been extensively studied so that it may be hard to find a good balance in summarizing previous work, however, there are a few review books/papers/reports that summarize many works (Whitehouse, 2009; Ekman, 2010; Steffen & Wu, 2011). These should be the backbones for another paragraph between the two on a brief overview of GIA (investigations) in Fennoscandia.

L32f: I suggest remove "that are tectonically quiescent". They are thought to be but there was and is more activity than quiescence, see e.g., Lindblom et al. (2015) and Lund et al. (2017).

L43: introduce abbreviations

L50: I miss Müller et al. (2012) in the references, although they call it land uplift model, while mentioning the study by Zhao et al. (2012) does not seem to fit. They used GNSS data for determination of the subsurface structure.

L52: northern "Central" Europe

L61f: please add how many velocity results are taken from those two papers, respectively. Did you use all stations from Kierulf et al. (2014)? Note that especially the many in Norway have short time spans and thus their velocities should be used with care. I would have advised to use only those with at least 5 years of observations and your results in Fig. 8 (top) show differences for many Norwegian stations. Might be that these are the newer stations. Also note that Kierulf et al. point to possible neotectonics along the Norwegian coast emerging in the velocities which should be picked up in the discussion/conclusions.

Figure 1: the LGM margin is not correct in Denmark, Germany and Poland when compared to Figure 1 in Hughes et al. (2015); also mention that Iceland was glaciated but ice extent is not shown (or cut figure)

L80: Please provide an overview of the 31 common sites and their values. Which station shows the large difference?

L82: Which uncertainties from Kierulf et al. (2014) did you use? The ones from GAMIT/GLOBK are indeed very low but the authors also provide uncertainties from the time series analysis using CATS where a combination of white noise and flicker noise was assumed. The latter should be preferred in a modelling analysis.

Section 2.2: Did you add the degree-1 estimates for GRACE?

L94: There are quite large uncertainties in the higher degrees, especially from degree 60 and higher. Does the Wiener filter leave high-degree signals at all? If not much is left your spatial resolution is much lower.

L98: What about the aliasing effects from tides, see e.g., Ray et al. (2003)? Were these considered?

Section 2.3: I partially miss some information! How are the corrections calculated? Are they spatially variable in each region? What method is used to calculate the elastic

signal in terms of vertical deformation from a mass balance signals? What time steps are used when acceleration is included? What (earth) model (if any) is used to calculate the elastic signal from current ice melt? You explain the input and the result but all intermediate steps are missing!

L117: Please add that the model considers anthropogenic changes only (as in the conclusions). I wonder why you do not use a global hydrology model like WGHM, which appears to perform OK in northern Europe (see e.g., Wang et al., 2013).

L119: Please specify the glaciated regions, i.e. in Scandinavia. I suppose you do not consider whole Scandinavia as a glaciated region. I also wonder if Scandinavia has glaciers of 2x2 degrees grid size so that the hydrological model has such large gaps? Jostedal Glacier is the largest with 487 km2 – much smaller than a 2x2 degrees grid.

L124ff: Please state in the beginning that you use and discuss published estimates. When reading I had the impression you do all the modelling yourself.

Table 1: I would like to see the detailed contribution from anthropogenic hydrology and glaciers for each area.

L152: altimetry results are also corrected for a GIA effect from GIA models, and those models might be erroneous. It's a bit chasing your own tail. See also Tamisiea (2011).

L162: Please refer to Fig. 2c.

L171ff: This is the weakest part of your study. You are not satisfied with your results as you expect something different. So, you basically tune your corrections, which are an average of published studies, until you end up with a result that you consider reliable based on your expectations. But what if the expectations are wrong or the real situation is much more different from the expectations. Wouldn't it be better to adjust the uncertainty for GRACE based on the, as it appears, rather problematic corrections? It might be that you end up suggesting the GRACE results as not feasible for usage in the semi-empirical model development due to all these issues and likely large uncertainties which would also allow a large range of suitable a priori models. As a matter of fact, Fig. 6 and Table 2 clearly show that the best result is mainly constrained by GNSS data. In the combined solution D3 the GNSS data are much more weighted than GRACE. The contribution from GRACE is minor and much less reliable. Hence, the question arises if GRACE should be used at all in this study and if this point is one of the main conclusions of this study!

L196ff: I have difficulties to acknowledge such large corrections especially due to Greenland mass loss knowing that there is a plate boundary in between where already parts of the GIA signal are altered, see e.g. Klemann et al. (2008). Note that I do not question the value which I assume is based on a simple 1D elastic model. Is the value the upper bound or average of different models tested? What earth structure did you use to calculate the effect? Is it similar to the a priori model that best fits your observations?

L212: "mass loss" or "mass changes" or really "mass loss changes" (due to acceleration)?

L223: Just wonder why ICE-5G is used while the new ICE-6G_C is available for quite some time now.

L237ff: The second paragraph which comes without references. Please check Steffen & Wu (2011) for a list of Fennoscandian GIA model parameter and consider the studies by Zhao et al. (2012), Kierulf et al. (2014), Root et al. (2014), Schmidt et al. (2014 (they also use ANU as you do!)), Nordman et al. (2015) and Root et al. (2015 – for the Barents Sea). Please also refer to some literature for a few words on GIA models and Earth parameter for the British Isles and the Barents Sea.

L237: Why do you use a fixed lithosphere and why 90 km? See references above which partly show quite different best-fitting values. Also, Steffen & Kaufmann (2005) point to differences in the lithosphere thickness in each of the regions you investigate (British Isles, Norwegian coast, Gulf of Bothnia, Barents Sea), further subdivision of (parts of)

northern Europe was done in Lambeck et al. (1998) and Steffen et al. (2014).

L239ff: Have a look into Nordman et al. (2015) who discuss this issue.

Table 2: Please check if the ratios are correctly calculated.

L329ff: There appears to be a misunderstanding and much information is misleading. It is important to go through the existing documentation (http://www.lantmateriet.se/globalassets/kartor-och-geografisk-information/gps-och-matning/referenssystem/landhojning/presentation-av-nkg2016lu.pdf). NKG2016LU is **not** a GIA model, it is as its name says a land uplift model. The observations are not corrected for any motion such elastic contributions from Greenland ice melt, hydrology, tectonics etc. The underlying GIA model is tuned to relative sea-level (RSL) data in northern Europe and GNSS data (with 80% weight on RSL data!), and is used as a gap filler in those areas where observations are not available, i.e. in the Baltic Sea. Hence, on land NKG2016LU represents the observed land motion - which has a very strong GIA component, of course. In addition, one should note that NKG2016LU is quite reliable in Fennoscandia (Norway, Sweden, Finland, Denmark) and performs well in the Baltic countries, but is not much reliable in Germany, Poland and eastern Europe as they are no or just a few observations (both for the semi-empirical and constraining the underlying GIA model). NKG2016LU largely relies here on the GIA model but which is tuned to give the best fit to the observations in Fennoscandia and the Baltic countries. The southern and eastern parts of the model are of less importance for the developers. As an interesting test, the NKG2016LU model could be on land treated as observation where the corrections of this study could be applied, and then used in the least-squares adjustment.

L353ff: Of course, the bias is 0.42 mm/a as NKG2016LU is the total observation where no elastic correction has been applied!

Section 3.4: I like this comparison as it nicely shows an important application. However, I wonder why you pick the North Sea where the GIA contribution is small. Would have

been nice to see how the model performs in the Baltic Sea and along the Norwegian coast. I also note that in the documentation of NKG2016LU a comparison to tide gauges has been made for Fennoscandia and the Baltic Sea.

Conclusion: I wonder what the best-fitting Earth structures are for you models. Are they like those used in the generation of ICE-5G and ANU? Although your model is discussed as a data-driven you should mention and discuss how much the contribution of the a priori models is in the final models. According to Table 2 it is quite large at the level of the GNSS data.

L419ff: What implications has this for the results of Root et al. (2015)?

Where can your model be downloaded?

L595-598: Please add the website path for the model, http://www.lantmateriet.se/sv/Kartor-och-geografisk-information/GPS-och-geodetisk-matning/Referenssystem/Landhojning/

**References (if not cited in the manuscript)**

Ekman, M., 2009. The Changing Level of the Baltic Sea during 300 Years: A Clue to Understanding the Earth. Summer Institute for Historical Geophysics, Åland Islands, Sweden, 155 pp.

Hughes, A.L.C., Gyllencreutz, R., Lohne, Ø.S., Mangerud, J., Svendsen, J.I., 2015. The last Eurasian ice sheets – a chronological database and time-slice reconstruction, DATED-1. Boreas 45, 1-45.

Johnston, P.J., Lambeck, K., 1999. Postglacial rebound and sea-level contributions to changes in the geoid and the Earth's rotation axis. Geophys. J. Int., 136, 537–558.

Klemann, V., Martinec, Z., Ivins, E.R., 2008. Glacial isostasy and plate motion. J. Geodyn. 46(3), 95-103.

Lambeck, K., Smither, C., Johnston, P., 1998. Sea-level change, glacial rebound and

mantle viscosity for northern Europe. Geophys. J. Int. 134, 102-144.

Lindblom, E., Lund, B., Tryggvason, A., Uski, M., Bodvarsson, R., Juhlin, C., Roberts, R., 2015. Microearthquakes illuminate the deep structure of the endglacial Parvie fault, northern Sweden. Geophys. J. Int. 201, 1704–1716.

Lund, B., Roberts, R., Smith, C.A., 2017. Review of paleo-, historical and current seismicity in Sweden and surrounding areas with implications for the seismic analysis underlying SKI report 92:3. Strålsäkerhetsmyndigheten report number 2017:35, ISSN: 2000-0456.

Müller, J., Naeimi, M., Gitlein, O., Timmen, L., Denker, H., 2012. A land uplift model in Fennoscandia combining GRACE and absolute gravimetry data. Phys. Chem. Earth 53-54, 54-60.

Nordman, M., Milne, G., Tarasov, L., 2015. Reappraisal of the Angerman River decay time estimate and its application to determine uncertainty in Earth viscosity structure. Geophys. J. Int. 201, 811–822.

Peltier, W.R., Andrews, J.T., 1976. Glacial–isostatic adjustment – I. The forward problem. Geophys. J. R. Astr. Soc. 46, 605–646.

Ray, R.D., Rowlands, D.D., Egbert, G.D., 2003. Tidal models in a new era of satellite gravimetry. Space Sci. Rev., 108(1–2), 271–282.

Root, B.C., van der Wal, W., Novak, P., Ebbing, J., Vermeersen, L.L.A., 2014. Glacial Isostatic Adjustment in the Static Gravity Field of Fennoscandia. J. Geophys. Res. 120 (1), 503-518.

Root, B.C., Tarasov, L., van der Wal, W., 2015. GRACE gravity observations constrain Weichselian ice thickness in the Barents Sea. Geophys. Res. Lett., 42, 3313–3320.

Schmidt P., Lund, B., Näslund, J-O., Fastook, J., 2014. Comparing a thermo-mechanical Weichselian Ice Sheet reconstruction to reconstructions based on the sea

level equation: aspects of ice configurations and glacial isostatic adjustment. Solid Earth, 5, 371-388.

Steffen, H., Kaufmann, G., 2005. Glacial isostatic adjustment of Scandinavia and northwestern Europe and the radial viscosity structure of the Earth's mantle. Geophys. J. Int. 163(2), 801-812.

Steffen, H., Wu, P., 2011. Glacial Isostatic Adjustment in Fennoscandia – a review of data and modeling. J. Geodyn. 52, 169–204.

Steffen, H., Kaufmann, G., Lampe, R., 2014. Lithosphere and upper-mantle structure of the southern Baltic Sea estimated from modelling relative sea level data with glacial isostatic adjustment. Solid Earth 5, 447–459.

Tamisiea, M., 2011. Ongoing glacial isostatic contributions to observations of sea level change. Geophys. J. Int. 186(3), 1036–1044.

Wang, H., Jia, L., Steffen, H., Wu, P., Jiang, L., Hsu, H., Xiang, L., Wang, Z., Hu, B., 2013. Increased water storage in North America and Scandinavia from GRACE gravity data. Nature Geosci. 6, 38-42.

Whitehouse, P., 2009. Glacial Isostatic Adjustment and Sea-level Change: State of the Art Report. TR-09–11, Svensk Kärnbränslehantering AB.

Wu, P., Peltier, W.R., 1982. Viscous gravitational relaxation. Geophys. J. R. Astr. Soc. 70, 435–486.

---

## Referee Comment (RC2) · Anonymous Referee #2 · 23 Mar 2018

This study provides a semi-empirical estimate with uncertainties of the glacial isostatic adjustment (GIA) signal for the northern Europe. The authors use a published methodology to adjust ("invert") an a priori set of GIA models using observations, bedrock uplift rates (GPS) and time-variable gravity (GRACE). The area coherently covered by this study is much larger than previous studies. So this study aims at providing a reliable GIA signal with robust uncertainties that can be used as correction for other kind of data (e.g. tide-gauges) over a large area of northern Europe.

This study deserves publication but I have few major concerns. 1) The correction of the data for the recent signal is calculated without considering its large variability over

the last two decades. More specifically GPS time spans are not uniform and, as I understand, the elastic correction is not computed for each station coherently with its time span. The elastic correction is not constant in time. Greenland mass loss for example has accelerated in the last decades.

2) As for the GRCE correction of the mass loss in Svalbard and the Russian arctic, the "large" discrepancies in Table 1 are mostly due to the different time spans and to the fact that the mass loss there has not been constant at all. I understand that from Cryosat is still hard to derive mass changes, so I wouldn't include the range of possible estimate. The most reliable estimates for Svalbard and Russian arctic come from ICESat and GRACE and over the same period they agree well enough. Since you need to extract a long term signal I would simply use the GRACE data over the period for which you have the most reliable corrections.

3) The re-scaling procedure of the mass loss in Svalbard and Russian is questionable and shows that the filter applied to the GRACE data is way too heavy. In fact Root et al. 2015 (doi.org/10.1002/2015GL063769) perform the same kind of correction on the GRACE data in the Barents Sea without the need to rescale. The authors also recognize that they cannot properly invert for the gravity data and that the initial filtering could have been too strong. So what if more a suitable filter were used on the GRACE data instead? How and how much would the result change? Is the gravity signature of the a priori GIA filtered with the same filter?

Minor comments

It is not explicitly said that is a semi-empirical study. It is called explicitly "inversion" which is quite misleading at first glance.

The use of the word "posterior": I suggest the use of "a posteriori" (if that is what the authors mean), but it is not necessary, it just sounds better to me.

L45-46. Forward models are supposed to have formal uncertainties only when the

models parameters are well (known and) constrained. The model parameters can have uncertainties depending on the error on the constraints (for the inversion). If a model parameter is unknown or have too large uncertainty then the error on the forward model is meaningless. The sentence is misleading (or incorrect), so I suggest rephrasing it.

L153-156. While this can be true, I think the GIA signal from LIA cannot explain large differences. The large differences come from computing the trend over different periods.

L241-244. The sentence is difficult to understand. Mostly because here the use of "... 'tuned' ice sheet history ..." is rather confusing. At first I believed it referred to the previous sentence so the following didn't make any sense. ICE5g and ANU for example are in fact 'tuned' ice histories. Anyway I believe the authors are referring to something else.

---

## Author Comment (AC1) · 6 May 2018

**Response to Reviewer 1.**

We thank the reviewer for his or her detailed comments on our manuscript. Below, we respond to the comments in turn.

**Comment 1:** Title: I have four issues, (1) Scandinavia is part of northern Europe but you miss to mention the Barents Sea, which is largely discussed in your manuscript, (2) abbreviations should not be used in the title except they are well known, (3) although I understand that you want to distinguish the GIA signal you investigate from current climate change-induced GIA signals, "long-term", i.e. its definition, is not the best word to me (see below), and (4) you estimate your signal also with help of GIA models. My suggestion would be: "The glacial isostatic adjustment signal in northern Europe, the British Isles and the Barents Sea estimated from satellite positioning and space gravimetry data, and geophysical modelling".

**Response 1:** We have modified the title to take into consideration several of the points the reviewer raises. We have removed 'Scandinavia' and 'long-term' from the title and replaced 'GIA' with 'glacial isostatic adjustment'. We agree in general that abbreviations should be introduced upon first usage, however, we think that both 'GPS' and 'GRACE' are well known and they both appear quite often as abbreviations in titles in similar literature. However, we change the title to refer to 'geodetic observations and geophysical models' – hopefully this is an acceptable compromise.

**Comment 2:** L11/L29-31: it seems "long-term" refers to "ice sheets: : : during the last glaciation" in your introductory lines 29-31. However, parts of this signal can result from previous glaciations, see e.g., Johnston & Lambeck (1999) and Root et al. (2015). This should be either specified or the word "long-term" be dropped.

**Response 2:** On line 11 in the abstract, and in the introductory paragraph, "long-term" has been removed. We have rewritten a couple of the sentences in the first paragraph to try to clarify the distinction between what we originally termed "long-term" (or paleo GIA) and GIA deformations from shorter-term/more recent processes. We were trying to make the distinction that although we are interested here in the present-day GIA response, that the signal has nothing to do with 'present-day' cryospheric change, such as climate-change enhanced mass losses of ice sheets and glaciers.

**Comment 3:** L12: suggest change GPS to GNSS and introduce abbreviation; GPS is one of the GNSSs like Galileo, GLONASS or BeiDou. You can specify in the main text that both Kierulf et al. and Blewitt et al. use GPS only.

**Response 3:** We have replaced GPS here with GNSS and defined the abbreviation. Rather than make a very long first sentence we have added a second sentence where the abbreviations are given explicitly.

**Comment 4:** L12: explain GRACE abbreviation

**Response 4:** We have also defined the GRACE abbreviation at first usage in the abstract. See also Response 3.

**Comment 5:** L12: delete "Scandinavia," (or do you mean northern Central Europe? – but my suggestion would be simply "northern Europe" which includes Scandinavia)

**Response 5:** Ok, 'Scandinavia' has been removed.

**Comment 6:** Introduction, i.e. L29-40: this is a rather short introduction that combines a paragraph without any references to a paragraph with references but already specifically focussed on the paper's topic. I suggest mention a few "early" general studies on GIA in the first paragraph, e.g. Peltier & Andrews (1976) and Wu & Peltier (1982). Otherwise it sounds that GIA should be well known for the reader. A reference for the 1 cm/a and the location should be added. I am aware that GIA in Fennoscandia has been extensively studied so that it may be hard to find a good balance in summarizing previous work, however, there are a few review books/papers/reports that summarize many works (Whitehouse, 2009; Ekman, 2010; Steffen & Wu, 2011). These should be the backbones for another paragraph between the two on a brief overview of GIA (investigations) in Fennoscandia.

**Response 6:** Regarding the overall structure: as a topic, GIA should be reasonably well known to the reader and we feel it is reasonable to have one general introduction paragraph followed by a second paragraph more focussed on our specific interests.

As for the references, we have added a couple of general references which were missing from the paragraph. We have also added Lidberg et al. (2010) and Kierulf et al. (2014) as two of the more recent references for the ~1 cm/yr of maximum uplift around the Gulf of Bothnia (which is already specified as the location, unless by location the GNSS station name is meant, but that might be overly specific for an introduction paragraph). As suggested, we add a paragraph in between the first and second paragraphs which summarizes some forward GIA modelling studies in the region, although we reiterate that forward modelling is not the focus of this paper.

**Comment 7:** L32f: I suggest remove "that are tectonically quiescent". They are thought to be but there was and is more activity than quiescence, see e.g., Lindblom et al. (2015) and Lund et al. (2017).

**Response 7:** Ok, this phrase is removed.

**Comment 8:** L43: introduce abbreviations

**Response 8:** We have explained the abbreviations here.

**Comment 9:** L50: I miss Müller et al. (2012) in the references, although they call it land uplift model, while mentioning the study by Zhao et al. (2012) does not seem to fit. They used GNSS data for determination of the subsurface structure.

**Comment 9:** Ok, we added the Müller et al. (2012) reference. We retain the Zhao et al. (2012) reference.

**Comment 10:** L52: northern "Central" Europe

**Response 10:** We change this sentence to refer to regions 'south of Scandinavia' and the British Isles, rather than northern or northern central Europe. It is unclear what clarification comes from adding 'central' here as some of the study area extends into the western and eastern parts of northern Europe.

**Comment 11:** L61f: please add how many velocity results are taken from those two papers, respectively. Did you use all stations from Kierulf et al. (2014)? Note that especially the many in Norway have short time spans and thus their velocities should be used with care. I would have advised to use only those with at least 5 years of observations and your results in Fig. 8 (top) show differences for many Norwegian stations. Might be that these are the newer stations. Also note that Kierulf et al. point to possible neotectonics along the Norwegian coast emerging in the velocities which should be picked up in the discussion/conclusions.

**Response 11:** We use 459 GPS velocities in total.

The Midas (Blewitt et al. 2016) data fill out the data coverage south of Scandinavia. Here, the data have been filtered to include only those data with time series duration of ≥10 years. The Midas data also have several sites that are located very close to each other. An additional filter is applied where all points that are within a 30 km radius of each other are collected, and the one with the largest number of usable data epochs is selected. The filtering for geography (i.e. south of the Kierulf dataset), time series duration, and spatial proximity to other sites leaves 309 Midas data points.

The remaining 150 velocities come from Kierulf et al. (2014), which is the full dataset provided in their supplementary material. The Kierulf et al. (2014) data (at least in the supplement from that paper) does not provide the length of the time series for the stations, so these data were not filtered (by us) for times series length. In Kierulf et al. (2014) and an earlier reference therein (Kierulf et al., Journal of Geodesy, 2012, doi:10.1007/s00190-012-0603-2), the authors indicate that although at least a 5 year time series would yield better precision, that they have opted to include data with time series durations of at least 3 years. If the time series duration information were provided we agree it would have been nice to try a cut-off of five years for this data set. Presumably the shorter time series data have larger associated uncertainties which will at least weight them less heavily in the solution.

In summary, the Kierulf et al. (2014) data have time series lengths of at least 3 years, and the Midas data that we have used have time series lengths of at least 10 years. We have added text that clarifies this in the first paragraph of the GPS section. We also add in the discussion/conclusion section a comment about the possible neotectonic signal in Norway mentioned by Kierulf et al. (2014).

**Comment 12:** Figure 1: the LGM margin is not correct in Denmark, Germany and Poland when compared to Figure 1 in Hughes et al. (2015); also mention that Iceland was glaciated but ice extent is not shown (or cut figure)

**Response 12:** Ok, we have changed the LGM margin in Figures 1 and 3 to be that of the 21 ka margin from Hughes et al. (2016), which indeed has a slightly different boundary across Denmark (the difference gets harder to distinguish across Germany and Poland). We also add to the caption that ice extent is not shown on Iceland.

**Comment 13:** L80: Please provide an overview of the 31 common sites and their values. Which station shows the large difference?

**Response 13:** We include a figure in the appendix that plots the 31 common sites. The site with the large difference is 'HONS' Honningsvaag in northern Norway. The values at station 'KOSG' at Kootwijk in the central Netherlands are also not the same within their uncertainties, although the difference there is smaller (0.194 mm/yr discrepant).

**Comment 14:** L82: Which uncertainties from Kierulf et al. (2014) did you use? The ones from GAMIT/GLOBK are indeed very low but the authors also provide uncertainties from the time series analysis using CATS where a combination of white noise and flicker noise was assumed. The latter should be preferred in a modelling analysis.

**Response 14:** It was indeed the latter uncertainties that were used (from the CATS analysis, so combination of white noise and flicker noise). This has been specified in the text.

**Comment 15:** Section 2.2: Did you add the degree-1 estimates for GRACE?

**Response 15:** We compute the trends in the CM frame. The GRACE data are already in the CM frame, so degree 1 coefficients are not necessary.

**Comment 16:** L94: There are quite large uncertainties in the higher degrees, especially from degree 60 and higher. Does the Wiener filter leave high-degree signals at all? If not much is left your spatial resolution is much lower.

**Response 16:** It is possible that the filter has removed some of the GIA signal, particularly at higher orders where there is more noise; the effective resolution is typically around 300 km (Siemens et al., 2013). We have added a sentence in the text here.

Siemes, C., Ditmar, P., Riva, R.E.M., Slobbe, D.C., Liu, X.L., & Hashemi Farahani, H. (2013). Estimation of mass change trends in the Earth's system on the basis of GRACE satellite data, with application to Greenland. Journal of Geodesy, 87(1), 69-87, doi:10.1007/s00190-012-0580-5.

**Comment 17:** L98: What about the aliasing effects from tides, see e.g., Ray et al. (2003)? Were these considered?

**Response 17:** In the level 2 processing of GRACE, several models remove the effects of (high-frequency) atmosphere and ocean signals on the gravity field estimation. In case of CSR release 5 tidal effects are removed with the use of the GOT4.8 model. Since the launch of GRACE in 2002, the removal of tidal aliasing signals became a standard protocol for all GRACE gravity field products. Therefore we did not include it in the description.

**Comment 18:** Section 2.3: I partially miss some information! How are the corrections calculated? Are they spatially variable in each region? What method is used to calculate the elastic signal in terms of vertical deformation from a mass balance signals? What time steps are used when acceleration is included? What (earth) model (if any) is used to calculate the elastic signal from current ice melt? You explain the input and the result but all intermediate steps are missing!

**Response 18:** The corrections are indeed spatially variable - this is clearly illustrated in Figures 2 and 3 which show the corrections in the study area. The time steps for the accelerations are annual. The solid Earth elastic signal is calculated (to degree and order 360) by applying the mass balance loads to a spherically symmetric Earth with PREM elastic parameters, computing a trend via least squares, and summing the regional contributions. We have added a couple of sentences in the section that explain this.

**Comment 19:** L117: Please add that the model considers anthropogenic changes only (as in the conclusions). I wonder why you do not use a global hydrology model like WGHM, which appears to perform OK in northern Europe (see e.g., Wang et al., 2013).

**Response 19:** We change the wording to indicate that the model considers anthropogenic changes only. In Wang et al. (2013) WGHM is shown to reproduce the peak hydrology signal well, but relative to the estimated separated signal, WGHM's peak signal is extended over a much larger area. In fact, the separated hydrology signal in this paper looks to be closer to ~0 mm/yr than it does the peak signal in much of the study area. Since hydrology models are in general still quite uncertain, and the separated hydrology signal is estimated to be small for much of the study area, it raises the question of whether improvement is achieved by applying a hydrology model at all. We err to the side of caution by only applying a small hydrological correction.

**Comment 20:** L119: Please specify the glaciated regions, i.e. in Scandinavia. I suppose you do not consider whole Scandinavia as a glaciated region. I also wonder if Scandinavia has glaciers of 2x2 degrees grid size so that the hydrological model has such large gaps? Jostedal Glacier is the largest with 487 km2 – much smaller than a 2x2 degrees grid.

**Response 20:** No, we do not consider the whole of Scandinavia as a glaciated region. The glaciers here are along the west coast of Norway, depicted by the white shading in Figure 1 (now indicated in the caption) and we have added a line in the text that indicates this. When we apply the correction here, the signal is filtered to be consistent with degree and order 96, the result being that the contributing signal from these glaciers is very small.

**Comment 21:** L124ff: Please state in the beginning that you use and discuss published estimates. When reading I had the impression you do all the modelling yourself.

**Response 21:** We add a line clarifying that the modelling comes from published estimates at the beginning (line 183). It is mentioned again in Table 1.

**Comment 22:** Table 1: I would like to see the detailed contribution from anthropogenic hydrology and glaciers for each area.

**Response 22:** This request is a bit unclear, since the contributions from the glaciated regions are already shown in Table 1. We describe in the text which of the summarized corrections we apply. It wouldn't be consistent to show the individual glaciers since when the mass loss signal is filtered to be consistent to degree and order 96 (same as the GRACE data) spatial resolution is lost. The combined contribution for hydrology and glacier mass loss is shown in Figure 2c. At any rate, the anthropogenic hydrology signal is quite small. In the Appendix, we have added a figure that separates the glacier and anthropogenic hydrology signals.

**Comment 23:** L152: altimetry results are also corrected for a GIA effect from GIA models, and those models might be erroneous. It's a bit chasing your own tail. See also Tamisiea (2011).

**Response 23:** The altimetry results correct for elevation changes due to GIA with a model, and of course there is always uncertainty associated with the application of any GIA model. Altimetry doesn't only measure cryospheric change, but can do so more reliably than GRACE since altimetry estimates are less sensitive to the GIA correction than GRACE. We have reworded the sentences here to make this clearer.

**Comment 24:** L162: Please refer to Fig. 2c.

**Response 24:** It is not appropriate to refer to Figure 2c here. The sentence on lines 161-162 state 'However, applying the averaged ice melt corrections to Svalbard and the Russian Arctic creates a large mass gain signal over these two areas and a relatively smaller signal in the central Barents Sea'. We do not show this effect in Figure 2c (or anywhere else); Figure 2c shows larger mass loss over Svalbard and the Russian Arctic than in the Barents Sea – this is expected because of present-day ice mass loss at those 2 locations. The large mass gain signal that is referred to in this sentence is why the ad-hoc filter (discussed below) was applied to the glacier mass loss correction in this region (otherwise it looks like Svalbard and the Russian Arctic gain mass relative to the Barents Sea).

**Comment 25:** L171ff: This is the weakest part of your study. You are not satisfied with your results as you expect something different. So, you basically tune your corrections, which are an average of published studies, until you end up with a result that you consider reliable based on your expectations. But what if the expectations are wrong or the real situation is much more different from the expectations. Wouldn't it be better to adjust the uncertainty for GRACE based on the, as it appears, rather problematic corrections? It might be that you end up suggesting the GRACE results as not feasible for usage in the semi-empirical model development due to all these issues and likely large uncertainties which would also allow a large range of suitable a priori models. As a matter of fact, Fig. 6 and Table 2 clearly show that the best result is mainly constrained by GNSS data. In the combined

solution D3 the GNSS data are much more weighted than GRACE. The contribution from GRACE is minor and much less reliable. Hence, the question arises if GRACE should be used at all in this study and if this point is one of the main conclusions of this study!

**Response 25:** This is a fair point, and it is one that we also make ourselves in the paper. It is not ideal to perform a tuning of the corrections to fit with expectations. We feel that it is likely that the real situation fits with our expectations in terms of the sign of the response, although the magnitude is indeed more difficult to constrain (i.e., we think that it is a reasonable to expect that when considering the paleo GIA signal there would be more mass gain in the region of the central Barents Ice Sheet than around its periphery). Note also that the VCE does adjust the uncertainty of the GRACE data by increasing it. It is still worth to try to use GRACE in this study, and its use has suggested future avenues that we could pursue to try to improve the reliability of the GRACE contribution. Also, just because the GRACE signal was problematic over the Barents Sea in this case, does not exclude the possibility that it can provide reliable constraint in other parts of the study area.

**Comment 26:** L196ff: I have difficulties to acknowledge such large corrections especially due to Greenland mass loss knowing that there is a plate boundary in between where already parts of the GIA signal are altered, see e.g. Klemann et al. (2008). Note that I do not question the value which I assume is based on a simple 1D elastic model. Is the value the upper bound or average of different models tested? What earth structure did you use to calculate the effect? Is it similar to the a priori model that best fits your observations?

**Response 26:** These are the values that are obtained for the solid Earth response using a 1D spherically layered model. It is an interesting comment that some of the elastic response may be altered across the plate boundary. If we understand Klemann et al. (2008), the horizontal velocities are much more sensitive to lateral variations than the vertical velocities (their Figure 8), and it is the vertical rates that we are correcting. At any rate, it is beyond the scope of our study to make an elastic correction using a laterally variable elastic model and it remains true that the 1D elastic models are commonly used to compute the vertical land motion and sea level response to present-day mass change scenarios both locally and globally.

The value of the elastic correction is the sum of the contributions from Greenland, Antarctica, glaciers and ice caps, and hydrology (described in Section 2.3). The elastic correction is not an average of different models. As stated in Section 2.3, the scenarios for Greenland and Antarctica are consistent with the results of Shepherd et al. (2012). The elastic earth model used to calculate the elastic correction is the Preliminary Reference Earth Model (Dziewonski and Anderson, 1981). All of the models in the a priori set also use PREM to describe the elastic structure of the Earth. Otherwise we are unsure what is meant by asking if the model is similar to the best fit model in the a priori set, because all models in the prior set are fully viscoelastic Earth models designed to model the paleo GIA response, whereas these corrections model only the elastic response to present-day load changes.

**Comment 27:** L212: "mass loss" or "mass changes" or really "mass loss changes" (due to acceleration)?

**Response 27:** We change the text in the caption here to read 'mass loss' (although the Greenland and Antarctic corrections do contain accelerations, see text).

**Comment 28:** L223: Just wonder why ICE-5G is used while the new ICE-6G_C is available for quite some time now.

**Response 28:** Yes, we could've used ICE-6G instead of ICE-5G. However, we also thought it equally interesting to compare the result of the data-driven model to that of a more recent forward model (ICE-6G). Therefore, using ICE-6G in the prior model set would compromise such a comparison because the data-driven results are to a degree dependent on the input model(s). There are two different ice sheet histories used and they are paired with a variety of viscosity profiles; together, the Earth and ice

model combinations should bracket a range of possible GIA signals for the region. We have added a sentence that explains this in Section 2.4.

**Comment 29:** L237ff: The second paragraph which comes without references. Please check Steffen & Wu (2011) for a list of Fennoscandian GIA model parameter and consider the studies by Zhao et al. (2012), Kierulf et al. (2014), Root et al. (2014), Schmidt et al. (2014 (they also use ANU as you do!)), Nordman et al. (2015) and Root et al. (2015 – for the Barents Sea). Please also refer to some literature for a few words on GIA models and Earth parameter for the British Isles and the Barents Sea.

**Response 29:** Although forward models are used in the prior input, this is not a forward modelling study. We are also not trying to infer specific values for classical GIA model parameters, we are trying to constrain the present-day GIA when a large range of plausible models are used as input, together with data. Of course, the range of possible parameters in the prior model set is informed by previous modelling studies. We have added a paragraph in Section 2.4 that summarizes some of the main results of these studies – the upper and lower mantle viscosity variations in our *a priori* set fit well with the min/max ranges inferred by these studies (see Response 30 for discussion of the lithospheric thickness).

**Comment 30:** L237: Why do you use a fixed lithosphere and why 90 km? See references above which partly show quite different best-fitting values. Also, Steffen & Kaufmann (2005) point to differences in the lithosphere thickness in each of the regions you investigate (British Isles, Norwegian coast, Gulf of Bothnia, Barents Sea), further subdivision of (parts of) northern Europe was done in Lambeck et al. (1998) and Steffen et al. (2014).

**Response 30:** For the study area as a whole, the lithospheric thickness may range from 71 – 160 km (see paragraph added to Section 2.4). The 90 km thickness falls within these value although we acknowledge the study could benefit from a wider use of lithosphere thickness values (a line has been added in the text indicating this). Note, however, that we are not trying to use our predictions to inversely infer Earth model parameters (which we now note in the introduction). If we were, having a range of lithospheric thickness values to choose from would indeed matter more; as it is, we are concerned with having a prior model set to input that contains a wide spread of possible GIA response values (the individual combinations that generate these variations matter less than the presence of the variations themselves). Nordman et al. (2015) as referred to below, also indicate that RSL data in central Fennoscandia cannot distinguish between lithospheric thicknesses (at least between the range of 46 – 146 km). And in general lower mantle viscosity is less well constrained than upper mantle viscosity, which makes the latter parameter probably the most important one.

**Comment 31:** L239ff: Have a look into Nordman et al. (2015) who discuss this issue.

**Response 31:** The text has now been edited here due to the added paragraph, but we guess either that the reviewer is referring to the range of mantle viscosity values used, or the idea of fitting an ice sheet model with a mantle viscosity profile. See Response 29 for a discussion of a range of mantle viscosity values. As for the fitting of the ice sheet model to a viscosity profile, ICE-5G is fit to a particular viscosity profile and the ice coverage in the ANU model is iteratively refined in conjunction with Earth model parameters; both ice sheet models are in their own way associated with best-fit viscosity values. In their study, Nordman et al. (2015) use well-constrained RSL data from Ångerman River in Sweden and investigate the fit of a large set of ice/Earth model combinations to the decay times of these data. They concluded similar to other studies that the RSL data are relatively insensitive to the ice sheet model. This may be true in Fennoscandia, but it doesn't necessarily apply in other parts of our study area (British Isles, Barents Sea) – so varying the ice sheet history for this study may still provide meaningful variation to the input model set.

**Comment 32:** Table 2: Please check if the ratios are correctly calculated.

**Response 32:** We calculated the ratios, and then rounded all of the numbers to two decimal points, hence the small discrepancy between the values in the table and their ratios. We have edited the ratios to be those generated by using the rounded values in the table.

**Comment 33:** L329ff: There appears to be a misunderstanding and much information is misleading. It is important to go through the existing documentation (http://www.lantmateriet.se/globalassets/kartor-och-geografisk-information/gps-ochmatning/referenssystem/landhojning/presentation-av-nkg2016lu.pdf). NKG2016LU is not a GIA model, it is as its name says a land uplift model. The observations are not corrected for any motion such elastic contributions from Greenland ice melt, hydrology, tectonics etc. The underlying GIA model is tuned to relative sea-level (RSL) data in northern Europe and GNSS data (with 80% weight on RSL data!), and is used as a gap filler in those areas where observations are not available, i.e. in the Baltic Sea. Hence, on land NKG2016LU represents the observed land motion - which has a very strong GIA component, of course. In addition, one should note that NKG2016LU is quite reliable in Fennoscandia (Norway, Sweden, Finland, Denmark) and performs well in the Baltic countries, but is not much reliable in Germany, Poland and eastern Europe as they are no or just a few observations (both for the semi-empirical and constraining the underlying GIA model). NKG2016LU largely relies here on the GIA model but which is tuned to give the best fit to the observations in Fennoscandia and the Baltic countries. The southern and eastern parts of the model are of less importance for the developers. As an interesting test, the NKG2016LU model could be on land treated as observation where the corrections of this study could be applied, and then used in the least-squares adjustment.

**Response 33:** We agree that the information as presented may be confusing, and we have reframed the discussion and figure to try to clarify the comparison.

The point here is to evaluate to what extent the presence/absence of an elastic correction in the GPS data influences the predicted model solution. We understand that the NKG2016LU model doesn't include a correction in the GNSS data for elastic/short-term signals. We start by explaining our own prediction from D1, which is a prediction, to the best extent that is possible, of the paleo GIA signal. When the model prediction is compared to the GPS data with the elastic correction, there is a bias of -0.01 mm/yr. Conversely, when the model prediction is compared to the GPS data without an elastic correction, there is a bias of -0.35 mm/yr, which is logical since this is approximately consistent with the magnitude of the elastic correction applied in Scandinavia. If we perform the same comparison with the NKG2016LU model values, the uncorrected GPS data yield a bias of -0.06 mm/yr over Scandinavia, whereas the corrected GPS data yield an average overprediction of +0.42 mm/yr, which is again consistent with expectation given the magnitude of the elastic correction. We could have of course just done this comparison with the D1 model, but it can be helpful to also look at the work of other studies and see a consistent tendency. Maybe a more direct way of looking at this is to compare the NKG2016LU land uplift estimate to the D1 estimate – when we do this, we see over Scandinavia that the average difference is +0.3 mm/yr. This difference is largely explained by the elastic correction on the GPS data applied in D1 – i.e., it is reflecting the difference between the total land uplift and the paleo GIA uplift. The magnitude of this difference may be larger than the uncertainty on the observations and/or the best-fit model prediction.

In the text of our conclusion, we write: *"The prediction of vertical land motion has a small but non-negligible sensitivity to the application of an elastic correction. … Therefore, the presence of such a difference in the vertical motion prediction suggests that while long-term GIA is the dominant contributor to vertical motion in central Scandinavia, that it is still worthwhile to correct GPS land motion rates for present-day elastic signals, so long as these signals are adequately approximated (e.g., Riva et al. 2017)."* This remains true for studies focussed on the paleo GIA signal, and was the essence of the point we were trying to make.

The reviewer also writes "The underlying GIA model is tuned to relative sea-level (RSL) data in northern Europe and GNSS data (with 80% weight on RSL data!)". It is not clear what the reviewer is trying to emphasize with this statement, but it makes sense that the underlying GIA model would also be based on RSL data, and it should be more heavily weighted towards those data since the GNSS data are also going into the semi-empirical solution. Although we haven't tried it, it is an interesting idea to take the NKG2016LU model as an 'observation', correct it for the elastic effect, and include it in

the inversion. However, it may also be a bit circular, since the GNSS data used to constrain both NKG2016LU and the models presented here are quite similar over Scandinavia.

**Comment 34:** L353ff: Of course, the bias is 0.42 mm/a as NKG2016LU is the total observation where no elastic correction has been applied!

**Response 34:** We are not surprised by this result either – that the difference is due to the presence/absence of the elastic correction is the point we were trying to make. The text has been reworded here now, but the point is that while the elastic correction is small, it can still cause a consistent difference between land uplift predictions and GIA predictions (see Response 33).

**Comment 35:** Section 3.4: I like this comparison as it nicely shows an important application. However, I wonder why you pick the North Sea where the GIA contribution is small. Would have been nice to see how the model performs in the Baltic Sea and along the Norwegian coast. I also note that in the documentation of NKG2016LU a comparison to tide gauges has been made for Fennoscandia and the Baltic Sea.

**Response 35:** The GIA signal is generally small in the North Sea (except at sites Hirtshals and Tregde), although different forward GIA models predict both positive and negative rates of sea-level change in the central North Sea indicating there is still uncertainty here - this makes it an interesting place to evaluate the data-driven model.

Based partly on the reviewer's comments, we have modified this section a bit in the revision. For the North Sea, we now use the tide-gauge rates that were presented in Frederikse et al. (2016b, Geophysical Research Letters 43, doi:10.1002/2016GL070750). The main difference here is that the time span over which the trend is calculated is longer (1958-2014 compared to 1980-2013). The method used for calculating the trends over the shorter time span may not be well suited for determining lower frequency signals, and it appears that using the longer time span decreases the inter-station spread of the inferred non-GIA signals. Use of the longer time span also facilitates adding a similar comparison for the Norwegian coast, following the reviewer's suggestion. The sea level trends for the Norwegian coast are also taken from Frederikse et al. (2016b), so the time spans are consistent for the North Sea and Norwegian comparisons. We find that removing the GIA signal decreases much of the inter-station variability for both regions. There is also a difference when the results are compared to the ICE-6G model predictions at these tide gauge locations. We have updated the text and figures throughout Section 3.4 to reflect these changes.

**Comment 36:** Conclusion: I wonder what the best-fitting Earth structures are for you models. Are they like those used in the generation of ICE-5G and ANU? Although your model is discussed as a data-driven you should mention and discuss how much the contribution of the a priori models is in the final models. According to Table 2 it is quite large at the level of the GNSS data.

**Response 36:** Table 2 gives the results of the VCE analysis. In model D1, both the GNSS data and prior model information have their uncertainties somewhat scaled down. In model D3, the uncertainties of the GNSS data are almost unscaled (factor of 1.02) while the uncertainties of the prior model are scaled down (by a factor of 0.64). This does not necessarily mean that the prior model contributes more than the GNSS data, since particularly in the former load centre, the original uncertainties are significantly larger than the data uncertainties. We have added a couple of sentences in the text that explains this.

As for the best-fitting Earth structures, yes, we can examine the best-fit empirical model, and then see with which model or models in the *a priori* set it most closely corresponds. We point out again that it was not the main goal of this study to infer Earth structure, in fact, one of the expectations of using a data-driven approach is the minimization of the uncertainty associated with forward models. Nevertheless, when we compare the predicted data-driven prediction(s) to models in the *a priori* set, the suggested upper and lower mantle viscosity values of around 3-6 $\times 10^{20}$ Pa s, and 5-30 $\times 10^{21}$ Pa s, respectively (similar to other studies, the upper mantle viscosity is better constrained than the lower).

These Earth models have a 90 km thick lithosphere, and as discussed in Response 30, if it were a primary goal of our study to infer these parameters, varying the lithospheric thickness would be useful. However, our upper and lower mantle viscosity inferences are quite consistent with those suggested by forward modelling studies whose main goal it is to constrain these parameters.

**Comment 37:** L419ff: What implications has this for the results of Root et al. (2015)?

**Response 37:** The study of Root et al. (2015) is a forward modelling study, whereas this is a semi-empirical modelling study. We believe that the GRACE signal we have used may have been aggressively filtered, so it is possible that some of the GIA signal has been lost. It is also the case that estimates of present-day mass loss in the Barents Sea vary over time period and estimation method, complicating the correction to GRACE for this effect. The outcome of the modelling study (for the D2 and D3 models) is thus dependent on the input GRACE signal and the assumed correction applied to it. The same is of course true for Root et al. (2015).

**Comment 38:** Where can your model be downloaded?

**Response 38:** We have placed gridded model predictions of vertical land motion and their uncertainties for the D1 model on the 4TU Centre for Research Data repository, https://data.4tu.nl/, doi:10.4121/uuid:4a495bbc-0478-483a-baef-19ff34103dd2. We have added this information at the end of the paper.

**Comment 39:** L595-598: Please add the website path for the model, http://www.lantmateriet.se/sv/Kartor-och-geografisk-information/GPS-och-geodetiskmatning/Referenssystem/Landhojning/

**Response 39:** We have added the link.

---

## Author Comment (AC2) · 6 May 2018

**Response to Reviewer 2.**

We thank the reviewer for his or her comments.

**Comment 1:**
1) The correction of the data for the recent signal is calculated without considering its large variability over the last two decades. More specifically GPS time spans are not uniform and, as I understand, the elastic correction is not computed for each station coherently with its time span. The elastic correction is not constant in time. Greenland mass loss for example has accelerated in the last decades.

**Response 1:** It is true that the elastic corrections for the GPS are not computed for each station within its time span – the trend is rather computed over the time interval 1993-2014. There is an acceleration for Greenland mass loss considered when the displacements for the elastic effect are computed (see text, Section 2.3). It is true that the computed elastic effect would be larger over shorter or more recent time spans. The full time span that we have used could therefore be considered a moderate elastic correction. It would be nice to try to compute the elastic corrections for each station for the appropriate time span – this would be possible for the data taken from Blewitt et al. (2016) but not possible with the Kierulf et al. (2014) dataset (as given) since the begin and end years of the trends were not provided. It is also this dataset that is centred over Scandinavia, where the elastic correction is largest.

**Comment 2:**
2) As for the GRCE correction of the mass loss in Svalbard and the Russian arctic, the "large" discrepancies in Table 1 are mostly due to the different time spans and to the fact that the mass loss there has not been constant at all. I understand that from Cryosat is still hard to derive mass changes, so I wouldn't include the range of possible estimate. The most reliable estimates for Svalbard and Russian arctic come from ICESat and GRACE and over the same period they agree well enough. Since you need to extract a long term signal I would simply use the GRACE data over the period for which you have the most reliable corrections.

**Response 2:** There are some large discrepancies in Table 1, even over the same time spans. For example, the glaciological estimates for Svalbard differ from each other over 2003-2009 and again from the IceSat and GRACE estimates over the same time period. That the mass loss has not been constant means that it is natural that the estimates for different time spans differ from each other. So it may be natural that the Cryosat estimate is larger than the IceSat estimate due to accelerated mass loss, and not due to unreliability of Cryosat estimations. If only IceSat is reliable, then this would mean that we stop the GRACE time series after 2009 and we prefer to use a longer time series.

**Comment 3:**
3) The re-scaling procedure of the mass loss in Svalbard and Russian is questionable and shows that the filter applied to the GRACE data is way too heavy. In fact Root et al. 2015 (doi.org/10.1002/2015GL063769) perform the same kind of correction on the GRACE data in the Barents Sea without the need to rescale. The authors also recognize that they cannot properly invert for the gravity data and that the initial filtering could have been too strong. So what if more a suitable filter were used on the GRACE data instead? How and how much would the result change? Is the gravity signature of the a priori GIA filtered with the same filter?

**Response 3:**
We have indicated that the treatment for mass loss in this region was problematic. Note that we applied altimetry-derived corrections, whereas Root et al. (2015) use a correction based on mascons which are smaller than the altimetry estimates. The *a priori* gravity information is unfiltered. The filter may have been too strong in the Barents Sea region, but less aggressive filters show comparable results over Scandinavia, so it is not clear to what extent the use of a different filter would result in a different prediction.

**Minor comments**

**Comment 4:** It is not explicitly said that is a semi-empirical study. It is called explicitly "inversion" which is quite misleading at first glance.

**Response 4:** In the introduction we now refer to the model as a semi-empirical model.

**Comment 5:** The use of the word "posterior": I suggest the use of "a posteriori" (if that is what the authors mean), but it is not necessary, it just sounds better to me.

**Response 5:** Ok, we have changed occurrences of posterior to a posteriori.

**Comment 6:** L45-46. Forward models are supposed to have formal uncertainties only when the models parameters are well (known and) constrained. The model parameters can have uncertainties depending on the error on the constraints (for the inversion). If a model parameter is unknown or have too large uncertainty then the error on the forward model is meaningless. The sentence is misleading (or incorrect), so I suggest rephrasing it.

**Response 6:** The inferred model parameters can have uncertainties that depend on the error on the constraints and the model uncertainties can be well or poorly constrained depending on the model's sensitivity to the data. What we meant also here is that GIA predictions themselves are often provided/discussed/used without uncertainties. The text has been reworded here: "The majority of GIA models are however forward models which can be limited by uncertainties in both the ice sheet model and Earth model. Furthermore, because a best-fit forward GIA model is generally a single Earth-ice model combination, their predictions of GIA deformations are typically provided without uncertainties".

**Comment 7:** L153-156. While this can be true, I think the GIA signal from LIA cannot explain large differences. The large differences come from computing the trend over different periods.

**Response 7:** We have suggested an LIA signal as a possibility, not as a certainty, as indeed there could be other explanations. That the GRACE signal differs from glaciological estimates and to a lesser extent altimetry estimates suggests that the GRACE signal may contain a solid Earth signal in addition to a mass loss signal (which would originate either from paleo GIA or LIA GIA, and spatially a signal from LIA would more likely be centred over the currently glaciated regions than the central Barents Sea region).

**Comment 8:** L241-244. The sentence is difficult to understand. Mostly because here the use of "... 'tuned' ice sheet history ..." is rather confusing. At first I believed it referred to the previous sentence so the following didn't make any sense. ICE5g and ANU for example are in fact 'tuned' ice histories. Anyway I believe the authors are referring to something else.

**Response 8:** Yes, we agree that the ICE-5G and ANU models are both in their way tuned ice sheet histories, and that is what we were referring to in this and the previous sentence. The text here has now been somewhat reworded, hopefully this clarifies the meaning. Our meaning was that if an ice sheet history is best fit with a particular viscosity profile then varying the viscosity profile over a wide range of values may make the predicted response variations larger than appropriate for a particular model; however, uncertainty in other parameters not considered would also likely make the uncertainties larger.